# Nutritional Value and Therapeutic Benefits of Dragon Fruit: A Comprehensive Review with Implications for Establishing Australian Industry Standards

**DOI:** 10.3390/molecules29235676

**Published:** 2024-11-30

**Authors:** Si-Yuan Chen, Cheng-Yuan Xu, Muhammad Sohail Mazhar, Mani Naiker

**Affiliations:** 1School of Health, Medical & Applied Sciences, Central Queensland University Australia, Bruce Hwy, North Rockhampton, QLD 4701, Australia; 2Research Institute for Northern Agriculture, Charles Darwin University, Ellengowan Drive, Brinkin, NT 0810, Australia; stephen.xu@cdu.edu.au (C.-Y.X.); muhammadsohail.mazhar@nt.gov.au (M.S.M.); 3Agriculture Branch, Department of Agriculture and Fisheries, Northern Territory Government, Darwin, NT 0828, Australia

**Keywords:** dragon fruit, nutritional value, therapeutic benefits, Australian market, industry standard

## Abstract

Dragon fruit, which is native to northern South America and Mexico, has become a significant crop in tropical and subtropical regions worldwide, including Vietnam, China, and Australia. The fruit (*Hylocereus* spp.) is rich in various bioactive phytochemical compounds, including phenolic acids, flavonoids, and pigments such as betalains and anthocyanins, which contribute to its antioxidant, anti-inflammatory, and anti-microbial properties. This comprehensive review introduces the origin, classification, and global production of dragon fruit, with a particular focus on its bioactive phytochemicals and therapeutic potential. Additionally, it critically evaluates the current industry standards for fresh dragon fruit production across key producing countries. While these standards primarily focus on quality, classification, and grading criteria, they lack focus on parameters related to the fruit’s bioactive content. The absence of established quality standards for fresh produce in the Australian dragon fruit industry presents a unique opportunity to develop guidelines that align with both international benchmarks and the therapeutic potential of the fruit. By addressing this gap, this review can potentially help Australia to position its dragon fruit industry to achieve greater consistency, competitiveness, and consumer appeal. As the demand for functional foods continues to rise, aligning Australian production practices with global standards becomes critical to meeting domestic market expectations. This review provides a comprehensive understanding of dragon fruit’s nutritional and therapeutic significance and highlights its potential role in establishing a robust standard for the Australian dragon fruit industry. A review of global industry standards reveled that Australian standard could incorporate classifications of dragon fruits, including external factors like appearance, size, and defect tolerance. Future research is needed to prioritize understanding of the impact of cultivation practices and environmental factors on the bioactive composition of dragon fruit, enabling the development of best practices for growers. Additionally, further studies are needed to evaluate the therapeutic effects of these bioactive properties through clinical trials, particularly their potential in preventing chronic diseases. The advancement of analytical methods for quantifying bioactive compounds will provide deeper insights into their health benefits and support the establishment of bioactive-oriented industry standards. Moreover, investigations of post-harvest handling and processing techniques could optimize the preservation of these valuable compounds, enhancing dragon fruit’s role as a functional food.

## 1. Introduction

Dragon fruit (*Hylocereus* spp.), commonly referred to as pitaya, originates from Mexico and northern South America and has grown into a globally cultivated crop due to its adaptability to tropical and subtropical climates [1]. Initially valued for its vibrant appearance and exotic appeal, dragon fruit has gained significant attention for its rich contents of bioactive phytochemicals such as flavonoids, phenols, anthocyanins, and betalains, which contribute to its health benefits. These bioactive compounds are known for their antioxidant, anti-inflammatory, anti-microbial, and anti-cancer properties, further elevating dragon fruit’s value as a functional food [2]. The commercial cultivation of dragon fruit has spread across key producing countries like Vietnam, China, and Indonesia, where favorable climatic conditions support its growth and large-scale production [3]. In contrast, Australia’s dragon fruit industry is relatively new, with potential for growth as interest in this exotic fruit increases domestically. Dragon fruit thrives in regions with annual rainfall between 25 and 50 inches and tolerates temperatures up to 40 °C, making it suitable for cultivation in Australia’s subtropical regions, particularly in Queensland and the Northern Territory [1].

Despite the growing global interest in dragon fruit as a functional food, research on its bioactive properties and therapeutic benefits, particularly within the context of Australian-grown varieties, remains sparse. This literature review seeks to address this gap by consolidating existing knowledge on dragon fruit’s nutritional and bioactive composition, as well as its therapeutic potential and the status of global production standards. Unlike major global producers, the Australian dragon fruit industry operates without established quality benchmarks, limiting its ability to align with international frameworks and meet the increasing demand for health-promoting produce. This review is novel in its emphasis on the implications of these gaps for the Australian industry and the potential to integrate bioactive-focused standards. By synthesizing existing findings, this work provides a foundation for the advancement of scientific understanding while supporting the development of industry standards that enhance the competitiveness and sustainability of Australian-grown dragon fruit. This comprehensive literature review explores the classification and bioactive phytochemical composition of dragon fruit, emphasizing its nutritional and therapeutic properties. Moreover, it highlights the global dragon fruit production landscape and the current global industry standards for fresh produce, underscoring the need for Australia’s emerging dragon fruit industry to align with international frameworks and capitalize on its competitive advantages by potentially establishing classifications for shape, size, defect tolerance, etc. This review aims to provide insights into the potential of establishing robust industry standards in Australia to enhance the marketability and sustainability of its dragon fruit production.

Published literature from the last two decades related to the cultivation, bioactive phytochemicals, nutritional value, therapeutic potential, and global industry standards for *Hylocereus* spp. (dragon fruit) was comprehensively reviewed. Relevant studies, including original research articles, review papers, and cross-referenced citations, were identified through numerous databases from the ‘EBSCOhost’ host platform, such as PubMed, SciFinder, Web of Knowledge, Scopus, and ScienceDirect. Search terms included combinations of (dragon fruit OR *Hylocereus* OR pitaya) AND (bioactive compounds, nutritional value, antioxidants, phenolic acids, flavonoids, betalains, anthocyanins, antimicrobial, anti-inflammatory, OR therapeutic properties). The inclusion criteria focused on articles that discussed the bioactive and therapeutic properties of dragon fruit and global production standards, with a specific emphasis on literature evaluating quality benchmarks in Australia and other major producing regions. Global industry standards and Australian dragon fruit market reports were sourced directly from government sites and horticultural bodies such as AgriFutures Australia.

## 2. Dragon Fruit Origin and Classification

Dragon fruit, also commonly referred to as pitaya (most commonly of the genus *Hylocereus*), belongs to the *Cactaceae* family, which comprises various cactus species. While this fruit is originally native to the arid regions of Mexico and northern South America, its commercial cultivation has spread widely beyond its native habitats. Nowadays, dragon fruit is extensively grown in tropical and subtropical regions around the world, including countries such as Vietnam, China, the Philippines, Israel, Australia, etc., where it has gradually become an important agricultural product [4].

The plant can thrive under specific environmental conditions, with ideal growth occurring in areas that receive annual rainfall between 30 and 50 inches. It is also highly tolerant of elevated temperatures, capable of withstanding heat up to 40 °C. As a member of the cactus family, dragon fruit adapts well to arid to semi-arid environments, making it particularly suited to regions with high light intensity and warm climates. However, while dragon fruit can survive in dry conditions, it requires a reliable supply of water to reach optimal growth. Additionally, the quality of the soil is a critical factor; the fruit flourishes best in fertile, well-drained soils that allow its root system to develop and support the plant’s growth. Thus, despite its cactus-like resilience to heat and light, adequate moisture and soil fertility are essential for the production of high-quality fruit [1]. This adaptation to diverse climates, coupled with its ability to thrive in a range of soil conditions, has made dragon fruit a viable crop in many regions outside its native range, contributing to its rising global popularity as both an agricultural product and a health-promoting fruit. 

*Hylocereus* spp. consists of several species, each displaying unique physical characteristics in terms of size, weight, and appearance (Figure 1). *Hylocereus undatus* (*H. undatus*) is one of the largest species in the group, with fruits ranging from 15 to 22 cm in length and weighing between 300 and 800 g. It is distinguished by its red peel contrasted with white flesh, giving it a visually appealing look [5]. This species is widely recognized in the dragon fruit market due to its large size and striking coloration. Another notable species is *Hylocereus polyrhizus* (*H. polyrhizus*), which produces smaller fruits, typically measuring 10 to 12 cm in length and weighing 130 to 350 g. The peel of *H. polyrhizus* is red, and its flesh is also red, differentiating it from the white-fleshed *H. undatus*. The presence of green at the top of the fruit remains a shared feature within the *Hylocereus* genus, contributing to its overall aesthetic appeal [5].

*Hylocereus purpusii* is similar in size to *H. polyrhizus*, with fruits ranging from 10 to 15 cm in length and weighing between 150 and 400 g. This species is unique for its light- or dark-red peel, with its red flesh creating a harmonious color scheme. Like other species in the genus, the green-tipped top adds visual consistency across the different types of *Hylocereus* fruit. *Hylocereus costaricensis* (*H. costaricensis*) is slightly larger and heavier, measuring 10 to 15 cm in length and weighing between 250 and 600 g. The peel is a vibrant red, while the flesh has a striking red–purple hue, making it distinct from the other red-fleshed varieties. The deeper color of the flesh further distinguishes *H. costaricensis* and adds to its visual and commercial appeal. Smaller species within the genus include *Hylocereus trigonus* (*H. trigonus*) and *Hylocereus megalanthus* (*H. megalanthus*), both of which produce fruits between 7 and 9 cm in length and weighing 120 to 250 g. Despite their similar sizes, these two species differ significantly in appearance. *H. trigonus* has white flesh, offering a contrast to the more commonly red-fleshed species, while *H. megalanthus* is recognizable by its yellow peel and white flesh, a combination that sets it apart visually from the other species in the *Hylocereus* genus [6].

## 3. Global Dragon Fruit Production

Major dragon fruit production regions are shown in Figure 2. These regions include China, Japan, Bangladesh, the Philippines, Vietnam, Australia, Indonesia, Sri Lanka, India, Thailand, South Africa, Spain, and the USA. Research on global dragon fruit production from 2017–2018 shows that Vietnam leads the world in dragon fruit production, with an impressive cultivation area of over 55,000 hectares and a total output of over 1,000,000 metric tons. Its productivity, ranging from 22 to 35 metric tons per hectare, highlights Vietnam’s favorable growing conditions [6]. As a result, Vietnam dominates the global market in terms of both volume and productivity, capitalizing on its large-scale operations. China follows closely, with 40,000 hectares of land dedicated to dragon fruit farming, yielding 700,000 metric tons. China’s productivity rate of 17.5 metric tons per hectare is slightly lower than Vietnam’s but still reflects a strong level of production efficiency [7]. Indonesia, despite having a smaller production area of around 8500 hectares, impressively produces over 200,000 metric tons of dragon fruit, with a productivity rate of 23.6 metric tons per hectare [7]. In contrast, Thailand operates on a more modest scale, with a cultivation area of nearly 3500 hectares and a total production of 26,000 metric tons. Its productivity, at 7.5 metric tons per hectare, is one of the lowest among the major producers [7]. Taiwan’s dragon fruit industry is relatively small, with a cultivation area close to 2500 hectares and an annual production of 49,000 metric tons. Although Taiwan’s production volume is lower than that of countries like Vietnam and Indonesia, its focus on maximizing yields (19.7 metric tons per hectare) from smaller areas, which ensures that its dragon fruit industry remains competitive [7]. Malaysia and the Philippines have cultivation areas of 680 hectares and 485 hectares, respectively. Malaysia produces 7820 metric tons, with a productivity of 11.5 metric tons per hectare, while the Philippines yields 6062 metric tons, with productivity ranging between 10 and 15 metric tons per hectare. Cambodia and India, with relatively small cultivation areas, produce 4840 metric tons and 4200 metric tons annually, respectively. Cambodia’s productivity stands at 11 metric tons per hectare, while India’s ranges from 8.0 to 10.5 metric tons per hectare, suggesting that both countries could benefit from improved farming practices to boost their output [7].

There is very limited literature on the Australian dragon fruit market, and it is still in its infancy, particularly when compared to larger global producers like Vietnam, China, and Indonesia. According to data from 2012–2015, Australia only produces 750 tons of fruit annually. As Australia’s dragon fruit industry is relatively new, there is significant potential for domestic growth. With increased investment in research and the expansion of cultivation areas, Australia could enhance its productivity and meet rising local demand. The country’s emphasis on sustainable farming practices and adherence to strict food safety standards gives it a competitive advantage, particularly in markets that prioritize ethically grown, premium-quality produce.

## 4. Nutritional Value

### 4.1. Chemical Composition

Red-fleshed (*H. polyrhizus*) and white-fleshed (*H. undatus*) dragon fruits are the cultivars found the most in the Australian industry, and as such, only these two species are discussed in this section. Due to the absence of literature on phytochemical concentrations and proximate analysis specific to Australian-grown dragon fruit, this review focuses on global data to provide a general understanding. Most of the nutrients present in *H. undatus* and *H. polyrhizus* are recorded at higher levels than those in popular tropical fruits, including jack fruit, pineapple, and mango (Table 1) [9,10,11,12,13,14,15,16,17,18,19,20]. It is worth noting that this Table 1 serves as a general comparison, as these values are highly influenced by factors such as geographical location, cultivation practices, and testing methodology [14].

The carbohydrate content in dragon fruit is similar to that of pineapple but lower when compared to other fruits, including banana, mango, and jack fruit. As a result, dragon fruit contains fewer calories, making it a suitable option for individuals aiming to lose weight [9]. Due to its relatively low sugar levels, dragon fruit is also considered a low-GI (glycemic index) fruit, beneficial for individuals with diabetes [21]. The main sugar present in dragon fruit is glucose (6 g/100 g), followed by fructose, and it also contains sorbitol, which contributes to its sweet flavor [14]. Additionally, dragon fruit is rich in vital minerals such as calcium, magnesium, potassium, iron, phosphorus, and sodium, all of which offer significant health benefits. These minerals play crucial roles in strengthening bones and teeth, regulating blood pressure and blood sugar, and aiding in the synthesis of essential amino acids [22].

Dragon fruit is also a valuable source of vitamins, including thiamine (vitamin B1, 2.4 µg/100 g), riboflavin (vitamin B2, 2.0 µg/100 g), niacin (vitamin B3, 12.6 µg/100 g), and ascorbic acid (vitamin C, 5.6 µg/100 g) [9]. With a high moisture content of over 85%, the fruit’s flesh is juicy and hydrating [23]. Its pH ranges from 4.5 to 5.0, classifying dragon fruit as a low-acid fruit, with malic acid being the predominant acid [14]. Table 1 also shows that *H. undatus* has lower amounts of fat, carbohydrates, energy, and total sugars than *H. polyrhizus*, making it a better choice among these two species for weight management and individuals with diabetes [9,21]. However, the red-fleshed dragon fruit is richer in fiber and minerals such as magnesium, phosphorus, and iron. In contrast, the white-fleshed dragon fruit has higher protein and potassium levels. *H. undatus* and *H. polyrhizus* contain comparable amounts of vitamins B1, B2, and B3 and vitamin C.

### 4.2. Bioactive Phytochemicals in Dragon Fruit

Phytochemicals, naturally occurring secondary metabolites found in various plant parts, are increasingly recognized for their significant bioactive properties and health benefits [24]. Recent studies have shown that dragon fruit is abundant in several important phytochemicals, including phenolic acids; flavonoids; and pigments such as carotenoids, betalains, anthocyanins, etc. These compounds are found in the flesh, seeds, and discarded peels of dragon fruit, which makes it a versatile source of bioactive components [25]. As detailed in Table 2, *Hylocereus* species, including *H. undatus* and *H. polyrhizus*, are particularly noted for their phytochemical richness, contributing to both their nutritional and medicinal value.

Among the most prominent phytochemicals in dragon fruit are hydroxybenzoic acids such as gallic acid, vanillic acid, syringic acid, and salicylic acid. Gallic acids, which can be found in the flesh, peel, and seeds, are well-known for their antioxidant, anti-obesity, and anti-diabetic properties [26,27,28,29]. Vanillic acid, which is present in the flesh and peel, exhibits antioxidant, anti-diabetic, anti-atherogenic, and anti-inflammatory effects [29,30,31,32,33]. Syringic acid, also found across the flesh, peel, and seeds, demonstrates antioxidant, anti-microbial, anti-cancer, and anti-diabetic potential [34,35]. Salicylic acid, although found only in the flesh, is primarily recognized for its antioxidant activity [9].

Dragon fruit is also a rich source of hydroxycinnamic acids, which include p-coumaric acid, caffeic acid, chlorogenic acid, and sinapic acid. These compounds are highly valued for their antioxidant and anti-diabetic properties. P-coumaric acid, found in the flesh, has demonstrated both antioxidant and anti-diabetic activities [31,36,37]. Caffeic acid, present in the flesh and peel, is primarily known for its antioxidant function [26,29,36]. Chlorogenic acid, located in the flesh, also contributes to anti-inflammatory and anti-diabetic effects [9,38], while sinapic acid, found in the flesh and seeds, further enhances antioxidant and anti-inflammatory responses [9,39].

Flavonoids are another key group of bioactive compounds in dragon fruit, including quercetin, catechin, rutin, phloridzin, and hesperidin. These flavonoids are abundant in the flesh, peel, and seeds and are recognized for their antioxidant, anti-diabetic, and anti-inflammatory properties. Quercetin, which is present in the flesh and peel of both *H. polyrhizus* and *H. undatus*, is known for its strong anti-inflammatory and antioxidant effects [40,41,42,43]. Catechin, also a potent antioxidant, has been detected in the flesh and peel [9,44]. Rutin, which is present in the flesh and seeds, has been linked to anti-inflammatory, antioxidant, and anti-diabetic effects [40,44,45]. Phloridzin and hesperidin, found in both the flesh and peel, contribute additional antioxidant and anti-diabetic properties, with hesperidin also displaying anti-cancer potential [46,47,48,49].

Carotenoids, particularly lycopene and β-carotene, are prominent in *H. polyrhizus*. Lycopene is known for its antioxidant and anti-cancer properties, while β-carotene offers antioxidant, anti-diabetic, and cardiovascular protective effects [50,51]. Additionally, anthocyanins such as cyanidin-3-glucoside, delphinidin-3-glucoside, and pelargonidin-3-glucoside are vital components of dragon fruit, particularly in *H. polyrhizus* and *H. undatus*. These anthocyanins found in the flesh and peel exhibit potent antioxidant and anti-inflammatory activities [40,52]. Lastly, betalains, which include betacyanin and betanin, are primarily concentrated in the flesh and peel of *H. polyrhizus*. Betacyanin is known for its extensive therapeutic properties, such as antioxidant, anti-microbial, anti-viral, and anti-inflammatory effects [53,54,55,56,57,58]. Betanin, which is well-known for its antioxidant and anti-microbial activities, is also present in the flesh and peel [55,56].

The diverse array of phytochemicals found in dragon fruit, including hydroxybenzoic acids, hydroxycinnamic acids, flavonoids, carotenoids, anthocyanins, and betalains, plays a crucial role in promoting various health benefits. These bioactive phytochemicals, distributed across the fruit’s flesh, peel, and seeds, underscore the potential of dragon fruit as a functional food with significant health benefits, including antioxidant, anti-inflammatory, anti-diabetic, and cardiovascular protective effects.
molecules-29-05676-t002_Table 2Table 2Bioactive phytochemicals isolated from dragon fruit (*H. polyrhizus and H. undatus*) and their health benefits.PhytochemicalStructureVarietiesPartsHealth BenefitsReferencesHydroxybenzoic acids




Gallic acid
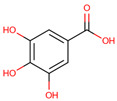
*H. polyrhizus* and *H. undatus*Flesh, peel, and seedsAntioxidant, anti-obesity, and anti-diabetes effects[26,27,28,29,37]Vanillic acid
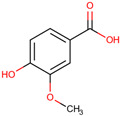
*H. polyrhizus* and *H. undatus*Flesh and peelAntioxidant, anti-diabetic, anti-atherogenic, and anti-inflammatory effects[29,30,31,32,33]Syringic acid
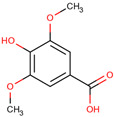
*H. polyrhizus* and *H. undatus*Flesh, peel, and seedsAntioxidant, anti-microbial, anti-cancer, anti-inflammatory, and anti-diabetes effects[34,35]Salicylic acid
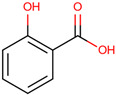
*H. polyrhizus* and *H. undatus*Flesh Antioxidant effects[9]Hydroxycinnamic acids




p-Coumaric acid
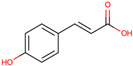
*H. polyrhizus* and *H. undatus*Flesh and peelAntioxidant and anti-diabetes effects[32,34,36]Caffeic acid
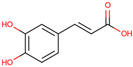
*H. polyrhizus* and *H. undatus*Flesh and peelAntioxidant effects[28,29,34]Chlorogenic acid
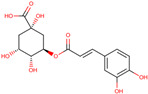
*H. polyrhizus* and *H. undatus*FleshAntioxidant, anti-inflammatory, and anti-diabetes effects[9,38]Sinapic acid
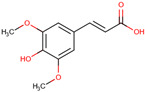
*H. polyrhizus* and *H. undatus*Flesh and seedsAntioxidant and anti-inflammatory effects[9,39]Flavonoids (Non-pigment)




Quercetin
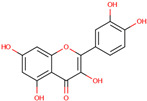
*H. polyrhizus* and *H. undatus*Flesh and peelAntioxidant, anti-diabetes, and anti-inflammatory capacities[40,41,42,43]Catechin
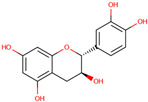
*H. polyrhizus* and *H. undatus*Flesh and peel Antioxidant effects[9,44]Rutin
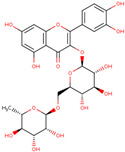
*H. polyrhizus* and *H. undatus*Flesh and seedsAntioxidant, anti-inflammatory, and anti-diabetes effects[43,45,59]Phloridzin
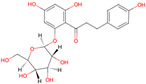
*H. polyrhizus* and *H. undatus*Flesh and peel Antioxidant, and anti-diabetes effects[48,49]Hesperidin
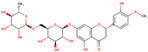
*H. polyrhizus* and *H. undatus*FleshAntioxidant and anti-cancer effects[46,47,48]Carotenoids




Lycopene
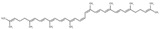
*H. polyrhizus*FleshAntioxidant, anti-cancer and anti-diabetes effects[50]β-carotene
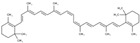
*H. polyrhizus*FleshAntioxidant, anti-diabetes, and anti-cardiovascular potential[50,51]Anthocyanins




Cyanidin-3-glucoside
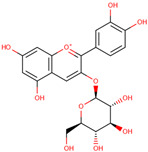
*H. polyrhizus*Flesh and peel (including *H. undatus*)Antioxidant and anti-inflammatory effects[43,52]Delphinidin-3-glucoside
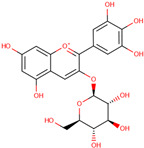
*H. polyrhizus*Flesh and peel (including *H. undatus*)Antioxidant and anti-inflammatory effects[43,52]Pelargonidin-3-glucoside
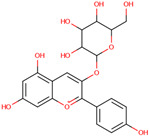
*H. polyrhizus*Flesh and peel (including *H. undatus*)Antioxidant and anti-inflammatory effects[43,52]Betalains




Betacyanin
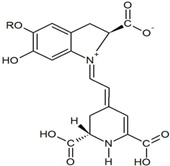
*H. polyrhizus*Flesh and peelAntioxidant, anti-microbial, anti-viral, and anti-inflammatory effects[53,54,55,56,57,58]Betanin
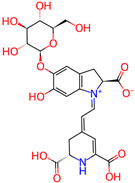
*H. polyrhizus*Flesh and peelAntioxidant and anti-microbial capacities[57,58]Note: Concentration data are not included due to significant variability across studies influenced by environmental and agricultural factors and the lack of data specific to Australian-grown dragon fruit.


### 4.3. Health Benefits of Dragon Fruit

For centuries, the medicinal properties of dragon fruit have been well recognized, with historical evidence revealing that the Mayan civilization used both its fruit and flowers for a variety of health-related purposes. These included managing blood sugar levels, promoting wound healing, acting as a diuretic, and treating conditions such as dysentery and tumors. Additionally, the seeds and flowers were often used to prepare beverages intended to alleviate gastritis, support bowel movements as a natural laxative, and enhance kidney function. The therapeutic benefits of dragon fruit are largely attributed to its rich concentrations of bioactive compounds predominantly found in the fruit itself. Notably, dragon fruit has been proven to be non-toxic in scientific studies. Animal models administered doses as high as 5000 mg/kg of dragon fruit extract exhibited no harmful effects or organ abnormalities, underscoring its safety for potential medicinal applications [2,30,60,61]. This favorable safety profile, coupled with its extensive bioactive composition, makes dragon fruit a promising candidate for further exploration in the field of natural medicine. The subsequent sections delve into the various health benefits attributed to dragon fruit, including its antioxidant, anti-inflammatory, anti-microbial, anti-diabetic, anti-cancer, anti-lipidemic/anti-obesity, anti-viral, and probiotic properties. These effects are supported by its bioactive compounds, which contribute to the fruit’s diverse medicinal potential.

#### 4.3.1. Antioxidant Capacities

The production of reactive oxygen species (ROS), commonly known as free radicals, is a key factor causing oxidative stress in the human body. These unstable molecules can cause extensive damage to essential cellular components, such as proteins, nucleic acids, lipids, and carbohydrates, potentially leading to mutations and the development of chronic diseases like cancer and cardiovascular disorders. Antioxidants are critical in counteracting this damage by neutralizing free radicals and reducing oxidative stress. Naturally occurring antioxidants, including carotenoids, flavonoids, tocopherol (vitamin E), and ascorbic acid (vitamin C), have been widely recognized for their strong free radical-scavenging capabilities [62].

Dragon fruit has emerged as a noteworthy source of antioxidants due to its high concentrations of polyphenols, carotenoids, and betalains, which are found in the peel, flesh, and seeds. The antioxidant activity of these phytochemicals is determined by their ability to neutralize free radicals by donating electrons, thereby preventing cellular and tissue damage [14]. Research conducted by Charoensiri et al. identified beta-carotene and lycopene as major antioxidants in dragon fruit, both of which are effective in neutralizing free radicals [63]. Additionally, phenolic compounds, particularly flavonoids, are regarded as significant contributors to the fruit’s overall antioxidant capacity and are highly concentrated in its edible parts [64]. Manihuruk et al. demonstrated the antioxidant activity of *H. polyrhizus* peel, attributing it to the abundance of phenolic compounds, hydroquinones, and flavonoids [65].

Further evidence from Tenore et al. supports these findings, as they used 2,2-diphenyl-1-picrylhydrazyl (DPPH) and ferric-reducing antioxidant power (FRAP) assays to compare the antioxidant activity between dragon fruit peel and flesh. Their results indicated that the peel extract exhibited significantly higher antioxidant activity than the flesh due to a higher concentration of betacyanin in the peel [32]. Similarly, studies using DPPH assays found that the antioxidant properties of *H. undatus* and *H. polyrhizus* are dose-dependent, with the red-fleshed variety demonstrating greater antioxidant activity due to its higher levels of phenolic compounds and betacyanin [66,67]. Zulkifli et al. further reported that elevated processing temperatures can enhance the antioxidant potential of the fruit [39].

Antioxidant properties in dragon fruit have been extensively studied across various plant parts, including the stem, leaves, flesh, peel, and seeds. The most abundant antioxidants in dragon fruit are phenolic compounds, betalains, and carotenoids, with alkaloids and vitamin C following in concentration [68]. This wide distribution of antioxidants contributes to the fruit’s overall bioactivity, making dragon fruit a promising functional food for reducing the risk of chronic diseases such as cancer. In vivo studies have demonstrated that dragon fruit extract assists in neutralizing free radicals produced during exercise. This is reflected in decreased levels of oxidative stress markers such as creatine kinase in animal models [69].

The antioxidant capacity of dragon fruit is closely linked to its phenolic content, which is influenced by external factors such as solvent extraction and temperature. For instance, a comparative study of white-fleshed dragon fruit peel revealed that methanolic extraction yielded a significantly higher phenolic concentration than chloroform, providing a concentration of 48.2 mg GAE/100 g, as methanol can extract both polar and non-polar compounds. Chloroform, being limited to non-polar compounds, resulted in lower antioxidant activity in DPPH assays [70]. Similarly, heat treatment ramping from 50 °C to 90 °C over one hour was shown to increase the antioxidant activities of dragon fruit, with almost double the total phenolic content determined in red-fleshed dragon fruit compared to the white-fleshed variety. Betacyanin, the primary antioxidant in red-fleshed dragon fruit, was identified as the main contributor to this enhanced activity, while the seeds also play an important role in maintaining antioxidant potential. Removal of the seeds significantly reduced the free radical-scavenging ability of the flesh [14]. Storage conditions also play a crucial role in preserving the antioxidant properties of dragon fruit. Research has shown that untreated dragon fruit experiences a slight increase in phenolic content during the first six days of storage, after which the content begins to decline. However, the addition of benzo (1,2,3)-thiadiazole-7-carbothioic acid S-methyl ester (BTH) to dragon fruit consistently exhibited increased phenolic and flavonoid contents, leading to enhanced antioxidant activity throughout storage [71]. These results emphasize the significance of optimizing extraction methods and storage conditions to preserve dragon fruit’s bioactive compounds and maximize its therapeutic potential.

In short, dragon fruit is a rich source of natural antioxidants, including polyphenols, carotenoids, and betalains, which contribute to its potent free radical-scavenging abilities. The antioxidant potential of dragon fruit is influenced by various factors, such as the extraction method, environmental conditions, and storage practices. As research continues to explore the fruit’s bioactive compounds, dragon fruit holds significant promise as a functional food capable of preventing oxidative stress-related diseases, supporting human health, and offering environmental resilience.

#### 4.3.2. Anti-Inflammatory Capacities

Dragon fruit possesses notable anti-inflammatory properties, as shown in Figure 3. This figure summarizes the main anti-inflammatory effects of *Hylocereus* spp., which are discussed in subsequent sections. Anti-inflammatory effects in dragon fruit are primarily due to the presence of bioactive phytochemicals such as anthocyanins, betalains, and squalene. Betalains, in particular, are recognized for their powerful antioxidant and anti-inflammatory effects. However, a significant challenge with betalains lies in their inherent instability under normal storage conditions. Environmental factors like oxygen, light, temperature, and pH greatly influence their stability, often leading to degradation and diminishing their effectiveness [25]. To overcome this issue, various strategies, such as encapsulating betalains in protective layers, have been proposed to enhance their stability and prolong their bioactive efficacy.

Rodriguez et al. derived betalains from the peel of red-fleshed dragon fruit and explored their anti-inflammatory effects. Betalains were encapsulated in maltodextrin matrices, combining them with pectin and gum arabic to form a protective barrier to enhance their stability. The effectiveness of both encapsulated and non-encapsulated betalains in reducing inflammation was evaluated using a duck embryo chorioallantoic membrane (CAM) vascular irritation assay. In this assay, sodium dodecyl sulfate (SDS), a known irritant, was used to induce vascular irritation. The results showed that encapsulated betalains were five to six times more effective in reducing vascular irritation compared to non-encapsulated forms. This improved anti-inflammatory activity was attributed to the increased stability of encapsulated betalains, which retained their free radical-scavenging capabilities—a key mechanism for reducing inflammation [73].

The anti-inflammatory action of betalains is closely tied to their antioxidant properties, as free radicals are major contributors to pro-inflammatory responses. By neutralizing these free radicals, betalains help to interrupt the inflammatory cascade, ultimately reducing tissue damage associated with chronic inflammation. Therefore, the ability of betalains to act as potent free radical scavengers plays a critical role in mitigating inflammation. Further research into the anti-inflammatory properties of the flesh and peel of dragon fruit, particularly white-fleshed *H. undatus*, has reinforced the bioactive potential of betalains. Eldeen et al. identified betalains as key contributors to the fruit’s radical-scavenging and anti-inflammatory capatities. Moreover, this research was the first to report the presence of squalene, a naturally occurring triterpene detected in the flesh of dragon fruit, with a concentration as high as 13.2%. Squalene exhibited considerable anti-inflammatory activity by inhibiting important pro-inflammatory enzymes, including acetylcholinesterase, cyclooxygenase-2 (COX-2), and 5-lipoxygenase (5-LOX), with of EC50 values up to 47 μg/mL. Since these enzymes are involved in regulating inflammation and neural functions, their inhibition suggests that squalene may have therapeutic potential for th treatment of inflammatory and neurodegenerative diseases [74].

The combined effects of betalains and squalene underscore the potential of dragon fruit as a natural anti-inflammatory agent, addressing both inflammation and related conditions, such as oxidative stress and neural disfunction. While betalains reduce inflammation by neutralizing free radicals, squalene strengthens this effect by inhibiting key pro-inflammatory enzymes. These findings indicate that dragon fruit, particularly when processed to preserve the stability of its bioactive compounds, holds promise as a therapeutic option for the management of inflammation, oxidative stress, and neural health.

#### 4.3.3. Anti-Microbial Potential

The rise of antibiotic resistance and the growing prevalence of infectious diseases have sparked a pressing need for alternative anti-microbial agents. Conventional antibiotics are becoming less effective due to the development of microbial resistance, prompting a shift towards the exploration of natural anti-microbials from fruits and vegetables, which are less likely to induce side effects or contribute to drug resistance in microbes [34]. Dragon fruit has shown great potential as a natural anti-microbial agent, displaying effectiveness against a wide range of microorganisms.

Various studies have demonstrated the anti-microbial effects of dragon fruit, particularly red-fleshed dragon fruit. One study assessed the ability of red dragon fruit to inhibit the growth of both Gram-negative and Gram-positive bacterial strains stored under cold conditions. The minimum inhibitory concentration (MIC) was measured by the broth microdilution method, with findings indicating that betacyanin levels in the fruit rose by almost 60% after 6 days of storage at 4 °C, followed by a gradual decrease. This increase was attributed to the hydrolysis of betalains into betalamic acid due to the fruit’s water content. The study found that dragon fruit stored under cold conditions exhibited enhanced anti-microbial activity compared to freshly harvested fruit, which was linked to its higher betacyanin levels at the six-day mark [58]. These findings suggest that proper post-harvest storage conditions could improve the fruit’s anti-microbial properties.

The anti-microbial properties of different parts of dragon fruit, including the peel, flesh, and whole fruit, have also been studied. When tested against 13 bacterial strains, two yeast strains, and four mold strains with the application of broth dilution, while whole fruit extracts were largely inactive, both the peel and flesh extracts exhibited significant anti-microbial activity. Polyphenolic compounds in the peel and flesh were identified as the primary contributors to this effect. Interestingly, Gram-positive bacteria were more susceptible to the extracts than Gram-negative bacteria, likely due to the additional outer membrane in Gram-negative bacteria, which acts as a barrier against anti-microbial agents [32].

Recent innovations have further enhanced the anti-microbial capacities of dragon fruit, particularly through the use of silver nanoparticles synthesized from dragon fruit peel extracts. These nanoparticles were tested for antibacterial activity against pathogens such as *Escherichia coli*, *Staphylococcus aureus*, and *Pseudomonas aeruginosa*, with gentamicin serving as the positive control. A disc diffusion assay revealed that the silver nanoparticles exhibited significant antibacterial activity against all the tested pathogens. The stability and increased anti-microbial efficacy of these nanoparticles were credited to the phytochemicals found in the peel extract of dragon fruit. [75]. The use of silver nanoparticles represents a promising method for boosting the anti-microbial efficacy of natural extracts.

Another study explored the antibacterial effects of the leaf extract of *H. undatus* and *H. polyrhizus* against bacteria that cause meningitis, including *Neisseria meningitidis*, *Listeria monocytogenes*, and *Streptococcus pneumoniae*. Both extracts demonstrated dose-dependent inhibition of these bacterial strains, with leaf extract of *H. polyrhizus* showing stronger antibacterial activity than *H. undatus*. This increased efficacy was linked to the presence of flavonoids, which are known to disrupt bacterial cell walls by forming complexes, thereby inhibiting bacterial growth [76]. These results suggest that dragon fruit may have specific applications in combating bacterial infections, including meningitis.

In short, these studies illustrate the broad-spectrum anti-microbial potential of dragon fruit, driven primarily by its bioactive phytochemicals, such as betalains, polyphenols, and flavonoids. The peel and flesh have been shown to exert strong antibacterial effects against a variety of microbial strains, while advanced techniques such as nanoparticle synthesis have opened up new approaches to enhance the effectiveness of these natural compounds. As antibiotic resistance continues to present a global health challenge, dragon fruit offers a promising natural alternative for the development of anti-microbial therapies that are both effective and less likely to contribute to resistance.

#### 4.3.4. Anti-Diabetic Capacities

Characterized by persistent hyperglycemia, diabetes mellitus (DM) is a common metabolic disorder caused by either the pancreas failing to produce enough insulin or the body becoming less responsive to insulin. This condition leads to elevated blood glucose levels, which can severely impact a patient’s overall health [77]. The dysfunction of insulin production or action is critical to the progression of DM. As the global incidence of diabetes continues to rise, the demand for effective treatment options has become urgent. Traditional medicines, particularly plant-based treatments, have attracted increasing attention due to their natural origins and reduced risks of side effects. Various medicinal plants have demonstrated promising results in managing diabetes by influencing key biochemical pathways, such as by inhibiting liver gluconeogenesis and improving cellular glucose uptake [78].

Among these natural therapies, the anti-diabetic properties of dragon fruit have attracted significant attention. Research indicates that both *H. undatus* and *H. polyrhizus* exhibit hypoglycemic effects that contribute to blood glucose regulation. For instance, Sudha et al. measured α-amylase inhibition activity of *H. undatus*, which is a critical mechanism for controlling diabetes. The study revealed that as the amount of dragon fruit extract added to enzyme wells increased (from 25 μL to 100 μL), α-amylase inhibition activity rose by 1% to 30%, highlighting the significant anti-diabetic potential of the fruit [79].

In addition to white-fleshed dragon fruit, red-fleshed varieties have also been studied for their anti-diabetic effects. Abd Hadi et al. conducted a study on type 2 diabetic patients, showing that consuming 400 g of red dragon fruit daily significantly lowered triglyceride levels, while consuming 600 g per day helped reduce hyperglycemia, decrease total and LDL (low-density lipoprotein) cholesterol levels, and increase HDL (high-density lipoprotein) cholesterol levels. These results emphasize red dragon fruit’s potential to regulate blood glucose in individuals with type 2 diabetes [80].

Further research into dragon fruit’s anti-diabetic properties has expanded understanding of its role in metabolic regulation. *H. polyrhizus* has been extensively studied for its effects on insulin resistance. Omidizadeh et al. investigated the impact of *H. polyrhizus* on insulin-resistant rats. This study involved feeding the rats both fresh and heat-processed dragon fruit samples with varying levels of phenolic contents and antioxidant activity. After six weeks, the rats exhibited a significant reduction in insulin resistance, alongside improved atherosclerotic changes and lower hypertriglyceridemia. The polyphenols, soluble dietary fiber and antioxidant compounds in dragon fruit were credited for these effects, suggesting that the fruit could be beneficial in reducing insulin resistance through dietary intervention [81].

Clinical trials on diabetic patients have further explored the effects of dragon fruit on lipid metabolism and glycemic control. A 7-week clinical trial conducted by Joshi et al. investigated the effects of *H. polyrhizus* consumption on blood glucose levels and lipid profiles in patients with type 2 diabetes. The study consisted of three phases—pre-treatment, treatment, and post-treatment phases—and found that consuming 400 g of dragon fruit daily significantly lowered triglyceride levels, while 600 g per day had a stronger effect in terms of reducing blood glucose levels. These results indicate that dragon fruit can positively influence both lipid metabolism and glycemic control in type 2 diabetes patients [80].

A study by Poolsup et al. also emphasized the hypoglycemic impacts of both *H. undatus* and *H. polyrhizus* in individuals with diabetes and pre-diabetes. The analysis revealed that dragon fruit was more effective at reducing blood glucose levels in pre-diabetic individuals, with greater reductions observed at higher consumption levels [82]. This dose-dependent trend suggests that increasing dragon fruit intake may be beneficial in managing early stages of diabetes or preventing the development of type 2 diabetes. The in vitro hypoglycemic activity of methanolic extracts of dragon fruit was also demonstrated by Ravichandran et al. using protein glycation inhibition assays. The polyphenolic content of dragon fruit was found to be a key factor in its anti-glycation and antioxidant effects [83].

Although increasing evidence supports the use of dragon fruit in the management of diabetes and its related complications, further research, particularly human clinical trials, is needed to confirm its efficacy. Most studies have focused on animal models or in vitro experiments, providing valuable insights, but more clinical validation is necessary to determine dragon fruit’s therapeutic potential for diabetic patients. Future research should focus on establishing optimal consumption patterns, dosages, and long-term effects to provide clearer guidance on incorporating dragon fruit into diabetes management strategies. In conclusion, dragon fruit shows considerable promise as a natural anti-diabetic agent. Dragon fruit’s potential to enhance insulin sensitivity, regulate blood glucose levels, and manage lipid profiles positions it as a promising candidate for diabetes treatment. However, additional human trials are necessary to thoroughly understand its mechanisms and confirm its therapeutic benefits in clinical applications.

#### 4.3.5. Anti-Cancer Potential

Imbalance between reactive oxygen species (ROS) and antioxidants in the body causes oxidative stress, a major contributor to the development of degenerative conditions such as cancer and Alzheimer’s disease. This stress damages cellular structures, promoting mutagenesis and uncontrolled cell growth. Rich in bioactive compounds such as polyphenols, betalains, tocopherols, and unsaturated fatty acids, dragon fruit has exhibited chemotherapeutic potential against various cancer cell lines [34,84]. These bioactive components work synergistically to reduce oxidative stress and inhibit cancer cell proliferation.

Dragon fruit’s strong antiproliferative properties are largely attributed to its phenolic compounds. These compounds exhibit significant anti-cancer effects by targeting several biochemical pathways. Figure 4 shows the anti-cancer potential associated with *Hylocereus* spp. For instance, betalains, which are abundant in dragon fruit, have been found to inhibit lipid peroxidation and suppress cyclooxygenase (COX-1 and COX-2) enzymes, key players in inflammation and cancer progression. By disrupting these pathways, betalains help prevent the growth of tumor cells [85]. Le et al. highlighted the role of phenolic compounds in enhancing dragon fruit’s antiproliferative activity, suggesting that these compounds are critical to its anti-cancer effects [25].

Various studies have evaluated both cytotoxic and antiproliferative effects of dragon fruit against several cancer cell lines. For example, strong anti-cancer effects on colon and prostate cancer cells have been observed in both methanolic and aqueous extracts of dragon fruit, suggesting its potential as a natural therapeutic agent [45]. Additionally, research on white-fleshed dragon fruit (*H. undatus*) revealed that with increases in concentrations of dragon fruit extract, a significant decrease was observed in the viability of human hepatocellular carcinoma (HEPG-2) cells, underscoring the dose-dependent nature of its cytotoxic effects on liver cancer cells [86]. Another study conducted using a 3-(4,5-dimethylthiazol-2-yl)-5-(3-carboxymethoxyphenyl)-2-(4-sulfophenyl)-2H-tetrazolium (MTS) assay confirmed the strong cytotoxic effects of dichloromethane peel extracts, further highlighting dragon fruit’s potential in cancer treatment [87].

A comparative study examined the effects of *H. polyrhizus* and *H. undatus* peel extracts on human breast, prostate, and gastric cancer cell lines, revealing dose-dependent cytotoxicity. The results showed that *H. polyrhizus* exhibited stronger inhibition, especially against human gastric cancer (MGC-803) cells, with inhibition rates exceeding 60% across all tested cell lines [67]. The superior anti-cancer activity of *H. polyrhizus* was attributed to its higher concentrations of pentacyclic triterpenoids and steroids, which are believed to contribute to its greater efficacy in inhibiting cancer cells [84].

In addition to these compounds, other phytochemicals in dragon fruit, such as α-amyrin and β-amyrin, also exhibit anti-cancer and antiproliferative effects. However, α-amyrin was found to have relatively weaker effects compared to the other compounds when tested against breast, prostate, and gastric cancer cell lines [67]. Kim et al. further supported these findings, ranking the anti-cancer efficacy of different dragon fruit extracts, with white-fleshed dragon fruit peel showing the highest activity, followed by red-fleshed peel and red-fleshed dragon fruit flesh. The peel of *H. undatus* was found to possess chemotherapeutic compounds, including flavonoids and polyphenols, at concentrations three to five times greater than other sources, which contributed to its superior antiproliferative activity [88].

Nitric oxide (NO), a well-known factor in promoting tumor growth through angiogenesis, also contributes to the development of cancer. Jayakumar et al. demonstrated that the proliferation of MCF-7 breast cancer cells increased by more than 150% following treatment with sodium nitroprusside, a nitric oxide donor. However, pretreatment with *H. undatus* extract significantly reduced cell viability, with a 600 μg/mL dose inhibiting MCF-7 cell growth by over 80%. The study attributed this effect to the phenolic compounds in dragon fruit, which exhibited strong free radical-scavenging activity, suppressing tumor growth [89].

Dragon fruit has also been shown to mitigate the toxic side effects of conventional cancer treatments. In one study, Wistar rats treated with dragon fruit juice concentrate prior to cisplatin chemotherapy experienced significant protection against cisplatin-induced nephrotoxicity. This protective effect was linked to the high betanin content in the juice, which preserved renal function and reduced lipid peroxidation levels [90]. In recent years, dragon fruit has also been explored as a component in nanotechnology-based cancer therapies. Divakaran et al. synthesized dragon fruit extract-capped gold nanoparticles (DF-AuNPs) using a green method. These nanoparticles exhibited high levels of cytotoxic activity against MCF-7 breast cancer cells. The nanoparticles, measuring between 10 and 20 nm, selectively targeted MCF-7 cells, demonstrating dragon fruit extract’s potential as both a capping and reducing agent in nanoparticle synthesis. This innovative approach may open new avenues for more targeted and effective cancer treatments [91].

In summary, extensive research highlights dragon fruit’s antiproliferative and cytotoxic potential against various cancer cell lines. Its bioactive compounds—such as polyphenols, betalains, flavonoids, and triterpenoids—are especially abundant in the peel and contribute to its chemotherapeutic properties. Dragon fruit’s ability to suppress tumor growth, reduce chemotherapy side effects, and facilitate novel therapies like nanoparticle synthesis underscores its multifaceted role in cancer treatment. However, while promising, further in vivo studies and clinical trials are needed to fully validate dragon fruit’s therapeutic potential and ensure its safe and effective use in cancer treatment.

#### 4.3.6. Anti-Obesity Capacities

Dyslipidemia and obesity are two major risk factors for the development of cardiovascular diseases, as they both contribute to the onset of atherosclerosis and other vascular complications [92]. Given that cardiovascular diseases remain one of the leading causes of death worldwide, there has been increasing interest in using natural products as therapeutic agents to manage these conditions. Various fruits and plant-based products have been extensively studied for their potential to regulate lipid metabolism and assist with weight management.

Dragon fruit (*H. polyrhizus*) has shown great promise in reducing cardiovascular risks. A study conducted by Hernawati et al. examined the effects of the peel powder of *H. polyrhizus* on lipid profiles, including total cholesterol, HDL, LDL, and triglycerides, in hyperlipidemic male mice. The study followed a randomized design, where mice were fed a high-fat diet for 20 days to induce hyperlipidemia, followed by treatment with different doses of dragon fruit peel powder (50, 100, 150, and 200 mg/kg BW/day) for 30 days. Blood samples were collected to monitor lipid levels, and the results showed significant decreases in total cholesterol, LDL, and triglyceride levels, along with increases in HDL levels, particularly in the groups receiving higher doses. The peel’s high crude fiber content (nearly 70%), which includes both soluble and insoluble fibers, was believed to contribute to these lipid-lowering effects. Soluble fibers bind to bile acids and cholesterol, enhancing their excretion, while insoluble fibers reduce fatty acid synthesis in the liver. Additionally, dragon fruit peel contains tocotrienols and antioxidants, which help regulate lipid metabolism and improve insulin sensitivity [93]. Another study explored the anti-lipase activity of *H. undatus* juice. Lipase, an enzyme involved in fat digestion, was inhibited by dragon fruit juice in a concentration-dependent model, with inhibition rates ranging from 6% to 46%, as assessed by the Rhodamine agar plate method [79]. By inhibiting lipase, dragon fruit juice may reduce fat absorption, offering a natural means of managing obesity and promoting weight loss.

In addition to these lipid-lowering effects, red dragon fruit has been found to counteract dyslipidemia, hypertriglyceridemia, and atherosclerosis in animal models. A study conducted by Omidizadeh et al. showed that consuming red dragon fruit reduced atherosclerotic changes in rats fed a fructose-rich diet. This was attributed to the high levels of soluble dietary fibers, polyphenols, and antioxidants present in the fruit, which collectively improved lipid metabolism and reduced oxidative stress [81]. Polyphenols, in particular, act as antioxidants that neutralize free radicals and prevent lipid peroxidation, protecting against vascular damage and reducing cardiovascular risk.

Dragon fruit’s beneficial effects on lipid metabolism are closely linked to its nutrient-rich composition. The fruit is abundant in dietary fibers, antioxidants like tocotrienols, and bioactive polyphenols, all of which support healthier cholesterol levels and prevent fat accumulation. Furthermore, dragon fruit’s ability to improve insulin sensitivity may help mitigate the risk of type 2 diabetes, which is often associated with dyslipidemia and obesity. Its inhibitory effect on β-Hydroxy β-methylglutaryl-CoA (HMG-CoA) reductase, an enzyme critical for cholesterol synthesis, and its ability to bind bile acids, promoting cholesterol excretion, further underscore its role as a functional food for the management of metabolic health [93].

While these animal studies present promising results, further research in human clinical trials is necessary to confirm dragon fruit’s long-term effects on lipid metabolism and cardiovascular health. Future studies should establish proper consumption guidelines and explore the therapeutic benefits of dragon fruit in preventing obesity and dyslipidemia. Despite the need for more evidence in humans, current research points to dragon fruit’s considerable potential as a natural agent for improving metabolic health and preventing cardiovascular diseases.

Overall, dragon fruit offers significant promise in managing obesity and dyslipidemia due to its rich contents of dietary fibers, antioxidants, and polyphenols. Its ability to lower total cholesterol, LDL, triglycerides, and body weight while raising HDL levels highlights its multifaceted role in promoting cardiovascular health. Additionally, its capacity to inhibit lipase activity and improve insulin sensitivity positions dragon fruit as a promising natural intervention for the management of obesity and related metabolic disorders. Further studies should focus on validating these effects in human populations and determining optimal doses for therapeutic benefits.

#### 4.3.7. Anti-Viral Potential

Dragon fruit, particularly red-fleshed species, has drawn considerable attention for its anti-viral potential, primarily due to the presence of betacyanins, which are potent bioactive compounds. Several studies have explored the anti-viral potential of red dragon fruit, with promising results against various viral pathogens. Chang et al. conducted a key study that explored the anti-viral properties of betacyanins derived from *H. polyrhizus* in relation to the type 2 dengue virus (DENV-2), which is responsible for severe dengue fever cases. Dengue, which is a mosquito-borne illness, poses a serious global health threat, with DENV-2 being one of the most virulent strains. The study employed a plaque reduction neutralization test to evaluate how betacyanins could inhibit viral replication. Using a viral strain isolated from an infected patient, the researchers ranked the concentrations of betacyanins in red dragon fruit as follows: phyllocactin (highest), betanin, isobetanin, and hylocerenin (lowest). This study also revealed that the anti-viral effect was dose-dependent, showing an IC50 (half-maximal inhibitory concentration) of 125.8 μg/mL and a selectivity index (SI) of 5.3 μg/mL. These results highlight betacyanins’ potential as natural anti-viral agents, particularly against DENV-2 [94]. The study provided compelling evidence of the efficacy of red dragon fruit in reducing viral yield, suggesting its usefulness in combating dengue virus infections.

Red dragon fruit has also been explored for its anti-viral capacities against SARS-CoV-2 (severe acute respiratory syndrome coronavirus 2), the virus responsible for the COVID-19 (Coronavirus disease 2019) pandemic, in addition to its potential efficacy against dengue. A molecular docking study examined the interaction between betacyanins and critical SARS-CoV-2 targets, including the receptor-binding domain (RBD) of the spike protein and the major protease (Mpro). The study demonstrated strong interactions between betacyanins and these viral receptors, suggesting that these compounds could inhibit viral entry and replication. Various molecular interactions, such as hydrogen bonds and van der Waals forces, further supported the potential of betacyanins as therapeutic agents for SARS-CoV-2. These findings point to the possibility of using betacyanin-rich dragon fruit extracts as natural inhibitors of coronaviruses, opening new research approaches for the treatment and prevention of viral infections such as COVID-19 [95]. Moreover, betacyanins from red dragon fruit have shown anti-viral potential against other respiratory viruses, such as influenza. A study found that treatment with betacyanins significantly reduced virus titers in cells infected with influenza A, which is a common cause of seasonal flu and pandemics. This suggests that betacyanins could help lower viral loads and limit the spread of influenza within the host [96].

The body of research on dragon fruit underscores its strong anti-viral potential, particularly through the bioactive effects of betacyanins. These compounds appear to inhibit viral replication by interacting with viral receptors and possibly modulating the host’s immune response to infection. However, while these studies present promising in vitro and molecular docking results, additional research is necessary to gain a comprehensive understanding of the mechanisms of the anti-viral effects of betacyanins. Moving forward, studies should concentrate on in vivo models and clinical trials to evaluate the safety, effectiveness, and pharmacokinetics of betacyanins in the treatment of viral infections. Additionally, exploring the synergistic effects of betacyanins with other anti-viral agents could pave the way for combination therapies, maximizing the therapeutic potential of dragon fruit in combating viral diseases.

#### 4.3.8. Prebiotic and Probiotic Properties

The health of the gut microbiota is essential for maintaining digestive health, and imbalances in this microbial community can lead to various gastrointestinal issues, such as diarrhea, constipation, inflammatory bowel disease (IBD), and irritable bowel syndrome (IBS). Prebiotics and probiotics play a significant role in promoting gut health, with prebiotics serving as nourishment for beneficial bacteria and probiotics contributing by replenishing these microbes [97]. Dragon fruit, known for its rich prebiotic fiber content, has emerged as a promising natural remedy for supporting gut health and managing digestive disorders.

The primary carbohydrates in both *H. polyrhizus* and *H. undatus* are glucose, fructose, and oligosaccharides, with the latter being the most critical to dragon fruit’s prebiotic properties. These oligosaccharides resist hydrolysis by human α-amylase and artificial gastric juices, allowing them to pass through to the colon intact. In the colon, they stimulate the growth of beneficial bacteria, including *Lactobacillus* and *Bifidobacterium*, demonstrating clear prebiotic activity [98]. This ability to support beneficial bacteria suggests that dragon fruit can maintain a healthy gut microbiome by promoting the proliferation of these helpful microorganisms.

Several studies have explored the prebiotic effects of oligosaccharides in dragon fruit. For example, Dasaesamoh et al. found that the fecal fermentation of dragon fruit oligosaccharides increased the populations of *Bifidobacterium* and *Lactobacillus* while reducing harmful bacteria such as *Bacteroides* and *Clostridium*. The fermentation process also produced short-chain fatty acids (SCFAs) like lactic acid, acetic acid, propionic acid, and butyric acid, which play vital roles in gut health. These SCFAs provide energy to colonocytes and contribute to anti-inflammatory processes, improved gut barrier function, and better metabolic health [99]. The production of SCFAs underscores the significant prebiotic potential of dragon fruit in supporting gut health and regulating digestive function.

Animal studies have further explored dragon fruit’s prebiotic properties. Khuituan et al. conducted research on how oligosaccharides extracted from dragon fruit influence male ICR/Mlac mice. Results showed that these oligosaccharides increased transit time in the upper gastrointestinal tract while reducing total gut transit time. Additionally, they acted as both bulk-forming and stimulant laxatives by increasing fecal mass and enhancing gut motility, which is particularly beneficial for the management of conditions like constipation. These results exhibited that dragon fruit oligosaccharides offer both prebiotic benefits and therapeutic effects for gastrointestinal conditions [84,100].

The prebiotic properties and related compounds of both *H. polyrhizus* and *H. undatus* have been quantified using high-performance liquid chromatography (HPLC), with red dragon fruit containing slightly more oligosaccharides (90 g/kg) than white dragon fruit (86 g/kg). These oligosaccharides serve as a carbon source for beneficial bacteria. In one study, *Lactobacillus delbrueckii* experienced significant growth when cultivated with dragon fruit oligosaccharides, increasing from 9.0 × 10^7^ to 6.2 × 10^9^ cells/mL in less than 48 h. *Bifidobacterium bifidum* also showed considerable growth, further confirming dragon fruit’s effectiveness as a prebiotic [56]. These results highlight the role of dragon fruit in promoting a balanced gut microbiome and supporting digestive health. Khalili et al. further confirmed the prebiotic properties of dragon fruit, revealing that oligosaccharides were more concentrated in the flesh of both *H. polyrhizus* and *H. undatus* than in the peel. They noted that red dragon fruit exhibited slightly greater prebiotic effects than white dragon fruit. This emphasizes the importance of consuming the fruit’s flesh to fully benefit from its prebiotic properties [85].

The positive impact of dragon fruit on gut health has broader implications beyond digestion. A balanced gut microbiota plays a crucial role in supporting immune function, reducing systemic inflammation, and regulating metabolic processes. Dragon fruit’s ability to increase SCFA production—especially the production of butyric acid, which is linked to reduced inflammation and improved insulin sensitivity—suggests that it could help manage metabolic disorders like obesity and type 2 diabetes. Additionally, dragon fruit’s role in promoting a diverse and balanced microbiome may contribute to the prevention of gut dysbiosis, a condition associated with various diseases, including autoimmune disorders and mental health conditions.

## 5. Uses of Processed Dragon Fruit

The importance of dragon fruit has been recognized not only for its nutritional value but also in folklore and traditional medicine. The fruit can be consumed in its raw form or processed into a variety of products, including juice, wine, spreads, and desserts. In traditional herbal medicine, dragon fruit has been valued for its therapeutic properties, especially in promoting digestive health and combating various ailments. Despite its multifaceted benefits, dragon fruit remained largely underappreciated and undervalued for many years. However, recent scientific investigations have shifted this perspective, unveiling a wealth of evidence supporting the potential health benefits of dragon fruit. Research has highlighted its significance in addressing inflammation, cancer, and diabetes, alongside its potential as a natural colorant in food and cosmetics [101]. Furthermore, the peels of the fruit have been found to possess a strong capacity to absorb toxins, further contributing to its value in food and environmental safety [102,103].

### 5.1. Food and Cosmetic Industry

One of the most remarkable industrial uses of dragon fruit lies in its high betalain content. Betalains are natural, water-soluble pigments found abundantly in the fruit, particularly in the *H. polyrhizus* species. These pigments are valued for their ability to add vibrant colors to food products without affecting taste. In addition to their coloring properties, betalains are rich in proteins, fats, fibers, and antioxidants, making them a valuable component in both the food and health industries [104].

Dragon fruit peel has shown considerable potential as a natural thickening agent in the food industry. Studies demonstrated that at 5% concentrations, alcohol air residues derived from dragon fruit peels exhibited viscosity properties comparable to those found in commercial thickening agents, suggesting that dragon fruit peel could serve as an effective alternative in food product formulations [105]. Additionally, the cosmetics industry has increasingly adopted betalains from dragon fruit as a natural alternative to synthetic colorants and chemicals, which are often associated with skin irritations and allergic reactions. Betalains provide various skin benefits without the risk of adverse side effects, making them a popular choice in skincare formulations [106].

Another valuable component of dragon fruit is its seeds, which are rich in essential oils. Both *H. polyrhizus* and *H. undatus* seeds contain significant amounts of oil, ranging from 18.3% to 28.4%, with notable total tocopherol contents of 36.7 mg/100 g and 43.5 mg/100 g, respectively [27]. Research has demonstrated that dragon fruit seed oil is particularly high in linoleic acid, which is an omega-6 fatty acid, surpassing the concentrations found in other seed oils like canola, linseed, sesame, and grapevine oils. The oil from white-fleshed and red-fleshed dragon fruit seeds contain 50% essential fatty acids, with linoleic acid contributing 48.5% and linolenic acid contributing 1.5% of the oil composition [107]. Another study reported linoleic acid contents of 540 g/kg for *H. undatus* and 480 g/kg for *H. polyrhizus*, confirming the oil’s richness in functional lipids [108]. These findings highlight the potential of dragon fruit seed oil as a valuable source of essential oils, with applications in both nutritional and cosmetic products.

### 5.2. Others

Dragon fruit’s versatility extends across multiple industries, offering valuable applications in food, beverage, and dairy production. For example, fruit wine made from *H. polyrhizus* contains a rich array of aroma compounds, such as esters (66.2%), alcohols (18.2%), alkanes (4.3%), acids (5.9%), aldehydes (0.1%), olefins (0.1%), and other volatile substances (0.2%) [109]. This diverse composition of volatile compounds contributes to the wine’s unique flavor profile, making it an attractive product for the beverage industry.

In the dairy sector, the addition of flesh of white-fleshed and red-fleshed dragon fruit into yogurt has been shown to improve its nutritional qualities. Adding dragon fruit flesh enhances lactic acid and total phenol contents, as well as antioxidant activity, while also accelerating the milk fermentation process [110]. These enhancements not only boost the nutritional value of yogurt but also contribute to its classification as a functional food, offering additional health benefits.

In the baked goods industry, dragon fruit peel flour has been explored as a partial replacement for wheat flour in cookie recipes. When 15% dragon fruit peel flour was added to wheat flour, the resulting cookies showed increased ash and fiber contents, as well as higher carbohydrate levels, larger diameters, and a greater spread ratio [111]. This suggests that dragon fruit peel flour can improve the nutritional profile of baked goods, making it a valuable ingredient for the creation of healthier snack options.

Furthermore, red-fleshed dragon fruit peel powder has been proposed as a fat substitute in ice cream formulations, particularly for those seeking low-calorie alternatives. By reducing the fat content and adding dietary fiber, dragon fruit peel powder enhances the nutritional value of ice cream while lowering its caloric content [112]. This innovative use of dragon fruit peel demonstrates its potential in producing healthier, calorie-reduced food products.

## 6. Australian Dragon Fruit Industry

### 6.1. Overview

The Australian dragon fruit market is relatively small but steadily growing, with most production concentrated in Queensland, New South Wales, South Australia, and the Northern Territory. Dragon fruit, or *Hylocereus* spp., was introduced to Queensland in the 1970s, and cultivation has since spread to other regions of the country. The primary varieties grown in Australia are white-fleshed *H. undatus* and red-fleshed *H. polyrhizus*. However, despite being an economically important fruit with multiple uses, dragon fruit cultivation in Australia remains niche. It accounts for a relatively small portion of the nation’s horticultural output, yielding around 740 tons annually from approximately 40,000 plants [4]. The market for locally grown dragon fruit is further limited by high production costs, which make it difficult for Australian growers to compete on price with cheaper imported dragon fruit from Southeast Asia. Despite the growing interest in dragon fruit, there are still limited amounts of literature and data available specifically focused on the Australian dragon fruit industry. Much of the current research on dragon fruit centers around larger global producers such as Vietnam, Thailand, and other Southeast Asian countries. In contrast, the Australian industry remains under-researched, and there is a distinct lack of comprehensive studies detailing the market dynamics, consumer preferences, and trade patterns specific to Australia. This lack of detailed information poses challenges for local growers, who need accurate data to make informed decisions about production, marketing, and strategic planning. Without sufficient research and robust industry data, the Australian dragon fruit industry is left at a disadvantage when attempting to compete in both domestic and global markets [113].

### 6.2. Imported Dragon Fruit

A significant challenge faced by the Australian dragon fruit market is the increasing competition from imports, particularly from Vietnam. Since 2017, the influx of imported dragon fruit has led to a noticeable drop in the price of Australian-grown produce. Appendix A compares the monthly average prices of Australian-grown dragon fruit with those of imported dragon fruit over a six-year period from 2014 to 2020 [113]. The figure shows that the price of Australian-grown dragon fruit remained consistently higher than that of imported dragon fruit throughout the period. This price difference reflects the higher production costs in Australia, particularly labor and resource expenses, compared to lower-cost countries such as Vietnam. The monthly fluctuations in prices also reveal seasonal trends. The prices of Australian-grown dragon fruit typically peak during the off-peak seasons, when local supply is lower and imported fruit is more abundant. Appendix A illustrates a critical turning point in the Australian dragon fruit market, showing the monthly average prices of Australian-grown and imported dragon fruit both before and after 2017, which is when imports from Southeast Asia, particularly Vietnam, were first introduced, providing a clear comparison of trends over time [113]. The figure highlights a distinct drop in the price of Australian-grown dragon fruit following the commencement of imports in 2017. Before 2017, the prices of Australian-grown dragon fruit were relatively stable, with occasional seasonal fluctuations. However, after 2017, there was a notable downward trend in prices, reflecting the impact of cheaper imported fruit on the local market. The figure underscores the significant market disruption caused by the introduction of imports, which placed additional pricing pressure on local producers and contributed to the ongoing challenges faced by the Australian dragon fruit industry.

### 6.3. Current Issues

One of the most pressing issues in the Australian dragon fruit industry is the relatively high production costs compared to imported fruit. Labor and resource expenses are significantly higher in Australia, making it difficult for local farmers to produce dragon fruit at competitive prices [114]. Additionally, the introduction of cheaper imported fruit has created downward pressure on prices, which continues to impact the profitability of Australian-grown dragon fruit. Australian growers also face limited opportunities for export due to strict biosecurity regulations and high costs associated with exporting to international markets [115].

Moreover, there is a lack of research into off-peak production methods, such as controlled environment agriculture, which could help increase the availability of Australian dragon fruit throughout the year. Currently, Australian growers are limited to peak production seasons, making it harder to compete with year-round imported fruit. Technologies including light stimulation for flowering and temperature control during fruit development remain underdeveloped. Further investment in these areas is needed to boost domestic production during off-peak periods [113].

Despite the challenges, Australian-grown dragon fruit still holds a reputation for superior quality, which appeals to local consumers. This preference for Australian-grown produce is especially evident during the off-peak season (from October to April), when local fruit commands premium prices [113]. However, the market share of Australian-grown dragon fruit continues to shrink in the face of increasing competition from imports. Australian growers need to invest in strategies such as year-round production, targeted marketing campaigns, and improved post-harvest handling techniques to differentiate their fruit from cheaper imports.

## 7. Current Industry Standards Worldwide

Standards for fresh produce grades are commonly employed by government agencies and private entities during the procurement process to ensure the quality of fresh fruits and vegetables [116]. In addition to these government-mandated standards, many countries and regions also implement “voluntary standards” for various types of fresh produce. These voluntary standards allow companies to establish their own internal benchmarks, either to align with local government regulations or to meet specific market and export requirements [117]. These standards are particularly useful for businesses looking to enhance the consistency and quality of their products in both domestic and international markets.

A good example of such a voluntary standard in Australia is the *Pineapple Best Practice Manual* developed by the Queensland Department of Primary Industries and Fisheries. This manual is used by Golden Circle Ltd., Australia as a reference for Queensland pineapple growers, guiding them in achieving the best possible quality for both fresh and processed pineapple products [118]. The manual provides detailed recommendations for industry-wide quality benchmarks, such as the ideal levels of total soluble solids (TSS) in fresh Queensland-grown pineapples. For instance, the TSS level, which is a key indicator of sweetness and overall fruit quality, is recommended to be at least 16% during most times of the year, with a slightly reduced standard of 13% in winter months to account for seasonal variations. The adoption of such voluntary standards has proven to be an effective way for industries to regulate themselves and maintain high quality in their products. It also allows producers to remain flexible and responsive to market demands while ensuring compliance with local regulations. In the global marketplace, voluntary standards are often critical for exporters, as they help to guarantee that products meet the quality expectations of international buyers and regulators, facilitating smoother trade and fostering consumer confidence.

Dragon fruit is a globally traded fruit, with production primarily concentrated in Southeast Asia and Latin America. To ensure consistency in terms of quality, safety, and market competitiveness, various countries and regions have established specific standards for the classification, grading, and handling of fresh dragon fruit. These standards dictate key parameters such as fruit size, external appearance, and post-harvest treatments. Based on available information, there are four publicly available country-specific standards (Thailand, the Philippines, Malaysia, and Mexico), one region-specific standard (ASEAN, Southeast Asia), and one international standard (Codex Alimentarius), each addressing different aspects of fresh dragon fruit quality. Table 3 shows a detailed comparison of dragon fruit standards. The Mexican standard is not included due to the unavailability of an English version.

The Codex Alimentarius Standard for Dragon Fruit (CXS 237-2003) [119] establishes a framework for classifying and grading dragon fruit intended for fresh consumption in international trade. This standard divides dragon fruit into three classes: the Extra Class, Class I, and Class II, each reflecting different levels of quality based on external appearance and defect tolerance. Fruits categorized under the Extra Class must exhibit superior quality, with very minor superficial defects permissible, as long as they do not impact the fruit’s overall aesthetic or internal quality. Class I allows for slight defects in shape and skin, with defects covering no more than 1 cm^2^ of the surface, while Class II permits more noticeable defects, with an allowance for defects of up to 2 cm^2^. The Codex standard also classifies dragon fruit based on weight, with categories ranging from 110 g to over 700 g for both red and yellow varieties. A tolerance of 5–10% is allowed for deviations in size and quality, depending on the classification of the fruit [119].

Similarly, the ASEAN Standard for Dragon Fruit (ASEAN Stan 42:2015) [120] provides guidelines for commercial dragon fruit, particularly within the Southeast Asian nations, where dragon fruit is a significant agricultural export. This standard was modified from Malaysian Standard MS 2201:2008 Fresh Pitahaya—Specification (ICS:67.080.10) [123] and Thai Agricultural Standard TAS 25-2015 [121]. Similar to the Codex standard, the ASEAN standard categorizes dragon fruit into an Extra Class, Class I, and Class II, with specific defect allowances corresponding to each class. Extra Class fruits must maintain near-perfect external appearance, with only slight superficial defects allowed. In Class I, minor defects in shape and skin are permissible, provided they affect no more than 5% of the fruit’s surface area. Class II, which is reserved for fruit that does not meet higher quality thresholds, allows for defects on up to 10% of the surface area. The size grading in the ASEAN standard is also similar to that of the Codex standard, with weight categories ranging from 110 g to over 700 g. The tolerance for size and quality defects ranges between 5% and 10% depending on the class [120].

The Thai Agricultural Standard (TAS 25-2015) [121], which governs the quality of dragon fruit in Thailand, follows a similar classification system but with a few notable differences in weight categories and defect tolerances. Similar to the Codex and ASEAN standards, it defines three quality classes: an Extra Class, Class I, and Class II. The Thai standard defines slightly different size categories compared to the Codex and ASEAN standards, with dragon fruit ranging from 200 g to over 600 g in weight for red- and white-fleshed varieties. Similar to other standards, a 5–10% tolerance for size and quality is allowed depending on the class [121].

The Philippine National Standard (PNS/BAFPS 115:2013) [122] follows a similar structure to the Codex and ASEAN standards. Dragon fruits are classified into an Extra Class, Class I, and Class II based on their appearance and defect tolerances. Extra Class fruits must exhibit superior quality, while Class I allows for slight defects affecting up to 5% of the surface area. Class II permits more significant defects, covering up to 10% of the surface area. The size range is 110 g to over 700 g, and a 5–10% tolerance applies to size and quality [122].

The Malaysian Standard (MS 2201:2008) [123] for fresh dragon fruit is structured similarly to the other standards, categorizing fruits into an Extra Class, Class I, and Class II. Extra Class dragon fruit must be of exceptional quality and free from defects except for very slight superficial blemishes. Class I allows for minor defects, such as scratches or bruises, but limits these to a small area of the skin (up to 5%). Class II permits more pronounced defects, allowing blemishes to affect up to 10% of the surface area, similar to the Thai and Philippine standards. The size range under the Malaysian standard is also similar, with categories ranging from 110 g to over 700 g. A 5–10% tolerance for size and quality is allowed [123].

## 8. Recommendation for the Establishment of an Australian Dragon Fruit Standard

Establishing a robust Australian dragon fruit standard requires consideration of several factors, including bioactive phytochemicals and alignment with current global standards. The bioactive phytochemicals found in dragon fruit, such as phenolic acids, betalains, flavonoids, anthocyanins, and carotenoids, offer significant health benefits. These compounds, known for their antioxidant, anti-inflammatory, and anti-microbial properties, should play a key role in differentiating Australian dragon fruit in terms of quality and nutritional value. To reflect this, the Australian standard could incorporate classifications that recognize the nutritional quality of dragon fruits in addition to external factors like appearance and size. For example, red-fleshed varieties, which contain higher levels of antioxidants like betacyanins, could be assigned to premium categories based on their bioactive contents.

To ensure Australia remains competitive in the international market, the proposed standard should align with existing frameworks, such as the Codex Alimentarius (CXS 237-2003) [119], ASEAN Standard (ASEAN Stan 42:2015) [120], Thai Agricultural Standard (TAS 25-2015) [121], and Philippine National Standard (PNS/BAFPS 115:2013) [122], to establish classifications on size, shape, and defect tolerance. According to these standards, the Australian dragon fruit standard could use a three-tier classification system consisting of an Extra Class, Class I, and Class II. Extra Class fruits would be the highest quality, free of significant defects and imperfections, while Class I would allow for slight defects and Class II would accept more pronounced defects as long as the internal quality remains unaffected. This system would ensure compatibility with international trade requirements, facilitating smoother export processes for Australian dragon fruit.

Additionally, the size classification should follow the model used in existing global standards. Most standards classify dragon fruits into weight categories ranging from 110 g to over 700 g. While Thailand uses a slightly narrower range (200 g to over 600 g), adopting the broader size categories used in the Codex and ASEAN standards would make Australian dragon fruit more adaptable to various markets. Furthermore, the Australian standard could include a 5–10% tolerance for minor deviations in size and quality, in line with the allowances made by other international standards. This tolerance would ensure flexibility in meeting market demands while maintaining high-quality standards.

A unique aspect of the Australian dragon fruit standard could be the inclusion of bioactive phytochemical testing. Whereas existing standards focus predominantly on external quality, introducing optional testing for bioactive compounds like total phenolics, flavonoids, or betacyanins could elevate the status of Australian dragon fruit as a premium health product. This could create a new market segment where fruits with higher phytochemical contents are graded more favorably, promoting the health benefits of Australian-grown dragon fruit. For example, just as Brix measurements are used to indicate sweetness in some fruits, a similar system could be developed to quantify the antioxidant capacity or phytochemical concentration of dragon fruit. This would highlight the unique nutritional properties of Australian dragon fruit, especially when compared to imported varieties.

Furthermore, given Australia’s unique climate and environmental conditions, the standard could emphasize sustainability and environmental resilience. Dragon fruits grown in harsher climates are known to produce higher levels of protective antioxidants, potentially giving Australian-grown fruits an advantage in the market. The inclusion of sustainable practices in the production process could further enhance the marketability of Australian dragon fruit, especially in environmentally conscious markets.

In short, an Australian dragon fruit standard should not only mirror global standards in terms of classification, size grading, and defect tolerance but also set itself apart by incorporating the nutritional and bioactive properties of the fruit. By promoting these health benefits and emphasizing sustainability, Australia could establish itself as a leader in high-quality, nutrient-rich dragon fruit production, offering a unique product in both domestic and global markets.

## 9. Strengths and Limitations of This Review

This comprehensive review consolidates current knowledge on the nutritional value, bioactive phytochemicals, therapeutic potential, and industry standards of dragon fruit, with a unique emphasis on its implications for the emerging Australian market. A key strength lies in its holistic approach, integrating findings from diverse sources, including peer-reviewed articles, industry reports, and government publications. This enables a thorough understanding of dragon fruit’s potential as a functional food and highlights gaps in global and Australian production standards.

However, the review also has limitations. The reliance on available literature may have inadvertently excluded unpublished studies or recent industry data not accessible through standard academic platforms. Additionally, while this review highlights potential bioactive benefits, the findings are primarily based on in vitro and animal studies, necessitating further validation through human clinical trials. Furthermore, the review’s focus on Australian industry standards is constrained by the limited availability of local data, reflecting the nascent state of dragon fruit research and production in the region.

## 10. Conclusions

In conclusion, this comprehensive review on dragon fruit highlights the significant potential of this crop in both nutritional and commercial aspects. With its origins in Mexico and northern South America, dragon fruit cultivation has expanded across tropical and subtropical regions worldwide, including Australia. The fruit is valued not only for its exotic appeal but also for its rich contents of bioactive phytochemicals such as phenols, flavonoids, and carotenoids, which offer various health benefits. Dragon fruit has been shown to possess antioxidant, anti-inflammatory, anti-cancer, and anti-microbial properties, making it an important functional food.

The global dragon fruit market is currently dominated by countries like Vietnam, China, and Indonesia, where favorable growing conditions and large-scale agricultural practices have driven high productivity. However, Australia has the potential to carve out a niche market for high-quality, sustainably grown dragon fruit. The country’s adherence to strict food safety regulations can further enhance its competitiveness, particularly in premium markets. For the Australian industry to grow, establishing national standards for dragon fruit production that align with international practices is crucial, especially for general quality indicators such as shape, size, and defect tolerance. By incorporating bioactive phytochemical profiles and adapting global standards to local conditions, Australia can improve the quality and marketability of its dragon fruit. This will ensure consistency in production, enhance consumer confidence, and open new opportunities for export.

Future studies are encouraged to prioritize the exploration of the influences of cultivation practices, environmental factors, and post-harvest handling techniques on the bioactive composition of dragon fruit. Additionally, clinical trials are needed to validate the therapeutic effects of its bioactive compounds, particularly in preventing and managing chronic diseases such as diabetes, obesity, and cancer. The advancement of analytical methods for profiling and quantifying bioactive compounds will provide deeper insights into their health benefits, aiding in the establishment of bioactive-oriented industry standards. Furthermore, interdisciplinary research on innovative processing methods, such as encapsulation and preservation techniques, could enhance the stability and bioavailability of these compounds, expanding the applications of dragon fruit in functional foods and nutraceuticals. These efforts will not only strengthen the scientific understanding of dragon fruit but also support the development of industry standards that improve marketability and sustainability.

## Figures and Tables

**Figure 1 molecules-29-05676-f001:**
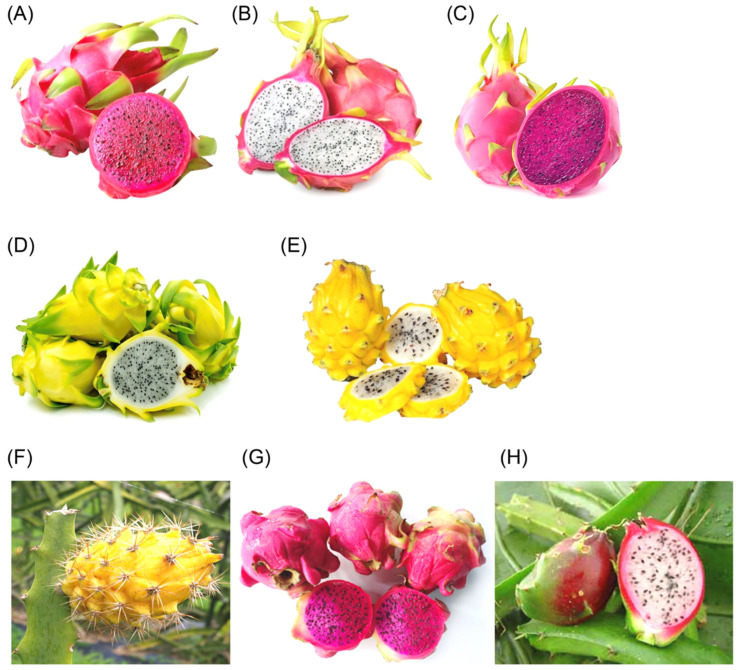
Different *Hylocereus* species [6]. (**A**) *Hylocereus polyrhizus*, (**B**) *Hylocereu. undatus*, (**C**) *Hylocereus costaricensis*, (**D**) *Hylocereus megalanthus*, (**E**) *Selenicereus megalanthus* (without spines), (**F**) *Selenicereus megalanthus* (with spines), (**G**) *Hylocereus purpusii*, and (**H**) *Hylocereus trigonus*.

**Figure 2 molecules-29-05676-f002:**
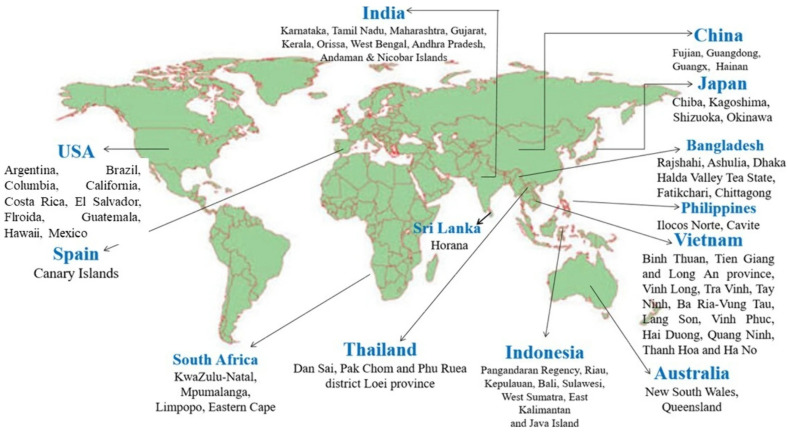
Map of major dragon fruit production regions [8].

**Figure 3 molecules-29-05676-f003:**
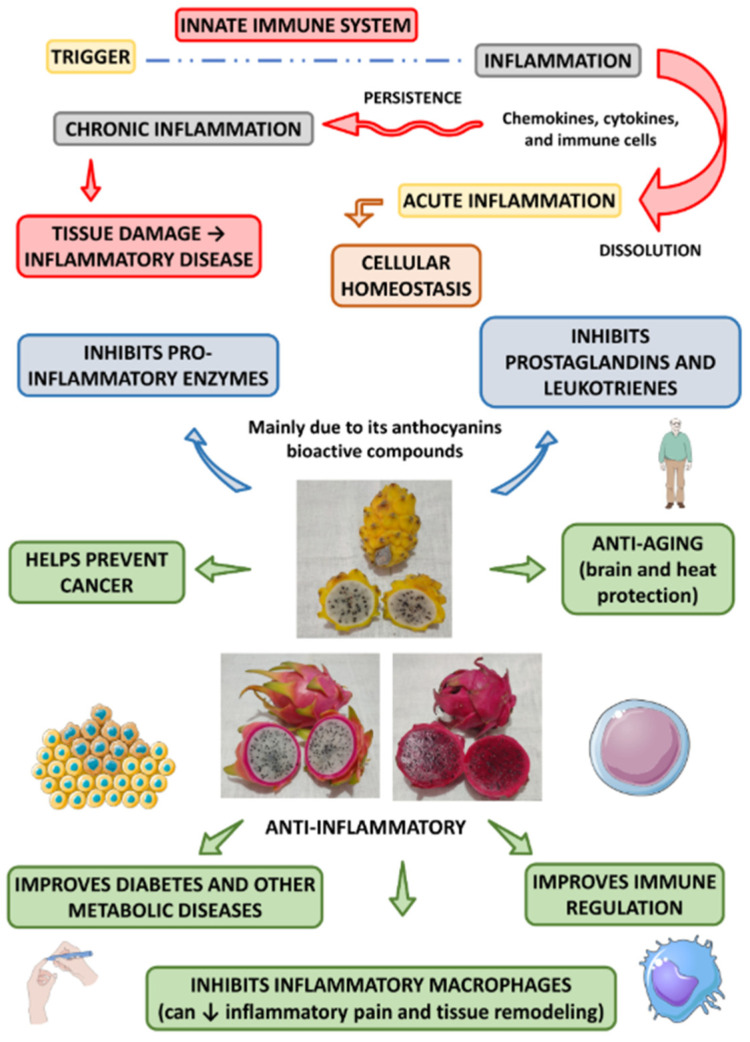
Major anti-inflammatory effects of *Hylocereus* spp. [72].

**Figure 4 molecules-29-05676-f004:**
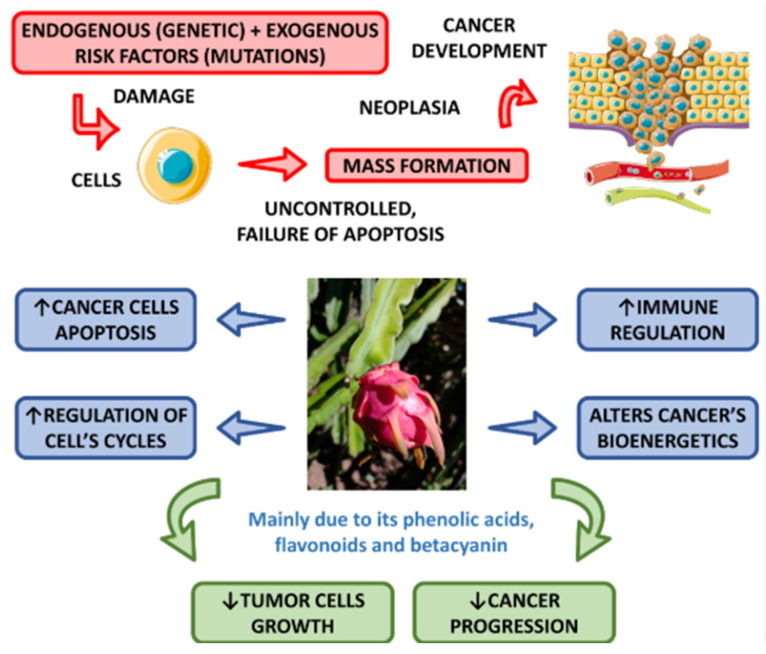
Major anti-cancer effects of *Hylocereus* spp. [72].

**Table 1 molecules-29-05676-t001:** Nutritional composition and recommended daily intake of dragon fruit (*Hylocereous* spp.) compared with other popular tropical fruits in Australia.

Composition	*H. undatus*	*H. polyrhizus*	Jack Fruit	Pineapple	Mango	Banana	RDI *
Protein (g)	0.5	1.1	1.9	0.6	0.4	1.1	64
Fat (g)	0.1	0.9	0.4	0.1	0.5	0.3	0.2
Carbohydrate (g)	9.5	11.2	25.4	11.8	15.0	22.8	-
Energy (KJ)	130	283	410	188.3	795	89	-
Fiber (g)	0.3	0.9	1.5	1.4	1.1	2.6	30
Total sugars (g)	8.6	9.2	20.6	8.3	13.7	12.2	-
Calcium (mg)	6.0	10.2	37.0	13	16	5	1000
Magnesium (mg)	26.6	38.9	27	12	19	27	420
Potassium (mg)	399.5	328.4	407	125	211	358	3800
Iron (mg)	0.4	3.4	1.1	0.3	0.4	0.3	8
Phosphorus (mg)	19	36.1	41.0	9	18	22	1000
Sodium (mg)	3.3	8.9	41.0	1	3	1	920
Vitamin B_1_ (μg)	2.2	2.4	90	78	40	80	1200
Vitamin B_2_ (μg)	2.0	1.3	400	29	70	2720	1300
Vitamin B_3_ (μg)	10.6	12.6	4000	106	1310	665	16,000
Vitamin C (mg)	5.6	4.4	10	16.9	92.8	8.7	45
References	[9,14,17,18]	[9,13,14,18]	[12,20]	[15]	[16]	[10,19]	[11]

Note: Results are expressed per 100 g fresh weight. Nutritional values may differ due to different geographical location, growing practices, and testing methodology. *: Recommended daily intake.

**Table 3 molecules-29-05676-t003:** Comparison of different dragon fruit standards.

Standard	Classifications	Defect Tolerance	Size Range (g)	Tolerances for Size and Quality	Reference
Codex (CXS 237-2003) [119]	Extra Class, Class I, and Class II	**Extra Class**: Very slight defects; **Class I**: Defects affecting up to 1 cm^2^; **Class II**: Defects affecting up to 2 cm^2^.	110–700+ (yellow and red/white varieties)	5–10% tolerance for size and quality deviations.	[119]
ASEAN (ASEAN Stan 42:2015) [120]	Extra Class, Class I, and Class II	**Extra Class**: Very slight defects; **Class I**: Defects affecting no more than 5% of the surface; **Class II**: Defects affecting up to 10% of the surface area.	110–700+ (red/white varieties)	5–10% tolerance for size and quality deviations.	[120]
Thailand (TAS 25-2015) [121]	Extra Class, Class I, and Class II	**Extra Class**: No defects; **Class I**: Defects affecting up to 5% of the surface; **Class II**: Defects affecting up to 10% of the surface.	200–600+ (red-/white-fleshed varieties)	5–10% tolerance for size and quality deviations.	[121]
Philippines (PNS/BAFPS 115:2013) [122]	Extra Class, Class I, and Class II	**Extra Class**: Very slight defects; **Class I**: Defects affecting no more than 5% of the surface; **Class II**: Defects affecting up to 10% of surface area.	110–700+ (red/white varieties)	5–10% tolerance for size and quality deviations.	[122]
Malaysia (MS 2201:2008) [123]	Extra Class, Class I, and Class II	**Extra Class**: Very slight superficial defects; **Class I**: Defects affecting no more than 5% of the surface; **Class II**: Defects affecting up to 10% of the surface area.	110–700+ (red/white varieties)	5–10% tolerance for size and quality deviations.	[123]

## Data Availability

Data sharing is not applicable to this article, as no datasheets were generated or analyzed during the current study.

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
