# Peer review of "Nutritional Value and Therapeutic Benefits of Dragon Fruit: A Comprehensive Review with Implications for Establishing Australian Industry Standards"

_molecules, 2024, doi:10.3390/molecules29235676_

Round 1
Reviewer 1 Report
Comments and Suggestions for Authors
In my point of view, the work conducted by Chen et al. is relevant and addresses an interesting topic. I believe that after some revisions it can be considered for publication in Molecules.
Dear authors, please mention the type of review in the title, in the abstract, and in the whole manuscript.
In the abstract, it is not clear your main results and the search strategy (databases, inclusion/exclusion criteria, etc…). Some directions and perspectives for future studies should be also given.
The Introduction section should be more elaborated. Please, provide the reasons for conducting your study, its novelty and what you expect to bring out to the population and scientific community with your work. Why is it important?
In section 3, please include a map of the worldwide regions where the dragon fruit is produced.
Section 4 is hard to follow, please include some tables and figures.
Before Conclusions, present the study’s strengths and limitations.
In the Conclusions, as I stated before some directions and perspectives for future studies should be also given. This has to be aligned with the abstract.
Author Response
|
Comments 1: Dear authors, please mention the type of review in the title, in the abstract, and in the whole manuscript.
|
|
Response 1: This is a general comprehensive review. This is now mentioned in the title, abstract, introduction and conclusion section. They can be found on page 1, line 3 and line 17; page 2, line 112 and page 28, line 1339, respectively, with track changes.
|
|
Comments 2: In the abstract, it is not clear your main results and the search strategy (databases, inclusion/exclusion criteria, etc…). Some directions and perspectives for future studies should be also given.
|
|
Response 2: As this is not a systematic or scope review, methodology is not necessarily needed. However, a small paragraph is now added in the introduction section, exhibiting the methodology. This can be found on page 2 from line 120 to line 133, with track changes. Recommendations on future research are now added to the abstract, on page 1 from line 30 to line 38, with track changes.
Comments 3: The Introduction section should be more elaborated. Please, provide the reasons for conducting your study, its novelty and what you expect to bring out to the population and scientific community with your work. Why is it important?
Response 3: Thank you for point this out. Extra information is now added to the introduction section, on page 2 from line 100 to line 112 with track changes.
Comments 4: In section 3, please include a map of the worldwide regions where the dragon fruit is produced.
Response 4: Map is now added to section 3, page 5, line 320, with track changes.
Comments 5: Section 4 is hard to follow, please include some tables and figures.
Response 5: Thank you for point this out. I understand that the information in section 4 is a bit overwhelming and thus I have added Figure 3 and Figure 4 on page 13, line 557 and page 17, line 808, respectively, with track changes, for better reading experience.
Comments 6: Before Conclusions, present the study’s strengths and limitations.
Response 6: This section is now added to the manuscript. Please refer to line 1323 to 1337 from page 27 to 28, with track changes.
Comments 7: In the Conclusions, as I stated before some directions and perspectives for future studies should be also given. This has to be aligned with the abstract.
Response 7: Extra information is now added to the conclusion section, from line 1358 on page 28 to line 1370, with track changes.
|

Reviewer 2 Report
Comments and Suggestions for Authors
1. **Abstract Revision**: The abstract needs to be revised as it currently does not clearly convey the opinions or information integrated in this review.
2. **Global Standards**: Please provide the requirements and standards for dragon fruit in different countries.
3. **Scope of Manuscript**: Is this manuscript focused solely on fresh dragon fruit? Consider adding sections on processing methods and processed products.
4. **Focus and Relevance**: The manuscript primarily introduces and organizes information about dragon fruit, with limited connection to Australia's management systems or standard establishment.
5. **International Comparisons**: Include a comparison of specifications between Australian dragon fruit and those from other countries.
Author Response
|
Comments 1: **Abstract Revision**: The abstract needs to be revised as it currently does not clearly convey the opinions or information integrated in this review. |
|
Response 1: Thank you for the feedback. The abstract is now revised to clearly convey the information. This changes can be found from line 13 to line 38 on page 1 with track changes.
|
|
Comments 2: **Global Standards**: Please provide the requirements and standards for dragon fruit in different countries.
|
|
Response 2: I have introduced “Current Industry Standard Worldwide” in section 7 (starts from line 1171 on page 24, this section concludes available industry standards for dragon fruit worldwide with a table to summarize and compare the differences (Table 3 starts from line 1229 on page 25).
Comments 3: **Scope of Manuscript**: Is this manuscript focused solely on fresh dragon fruit? Consider adding sections on processing methods and processed products.
Response 3: The primary focus of this manuscript is on fresh produce, particularly in the context of current industry standards for classification, quality, and grading criteria across major producing countries, as well as their relevance to the Australian dragon fruit industry (please refer to line 20, 22 on page 1 and line 115 in the introduction section on page 2). Adding sections on processed products would broaden the scope beyond the intended focus of the manuscript. The review aims to address the gaps in fresh dragon fruit production standards, particularly the absence of parameters related to bioactive content in global and Australian standards. This narrower focus allows us to provide a more detailed and meaningful analysis of fresh dragon fruit within the context of its bioactive compounds and therapeutic potential.
Comments 4: **Focus and Relevance**: The manuscript primarily introduces and organizes information about dragon fruit, with limited connection to Australia's management systems or standard establishment.
Response 4: As mentioned in the abstract (from line 22 to 30 on page 1) and introduction section (from line 100 to 112), there is no industry standard for dragon fruit in Australia currently available and no related management system related to it.
Comments 5: **International Comparisons**: Include a comparison of specifications between Australian dragon fruit and those from other countries.
Response 5: There is currently no published data on the proximate composition, phytochemical content, or specific quality specifications of Australian-grown dragon fruit. This reflects a significant gap in the literature, which we have noted in the manuscript as an area for future research (from line 30-38 on page 1, from line 336-338 on page 5 and from line 1358 to line 1370 on page 28). Additionally, the concentration of nutrients/bioactive compounds highly depends on growing environment, farming practices, testing method, etc., making direct comparisons challenging and potentially inconsistent. Table 1 from line 367 on page 6 serves as a general comparison between dragon fruit and other popular fruits. Its purpose is to provide readers with an overall understanding of how dragon fruit’s nutritional value compares to other fruits, rather than to focus on specific compositional differences between regions or growing conditions. |

Reviewer 3 Report
Comments and Suggestions for Authors
Dear Authors,
Thank you for this interesting study, Nutritional Value and Therapeutic Benefits of Dragon Fruit: A Review with Implications for Establishing Australian Industry Standards.
In general, the article is well-written and clearly articulates and highlights key points and topics. Specifically, the section “Recommendation on Australian Dragon Fruit Standard Establishment” enhances the novelty of this review by proposing a potential call to action for those involved in the production, processing, and consumption of dragon fruit.
Although the article is strong overall, the abstract is weak and needs improvement. Here are some specific comments:
I encourage the authors to clearly state the aim of this review instead of using vague phrases like “not only introduces” as mentioned in lines 17–19.
Line 22: The phrase “in contrast, the Australian” is grammatically unclear. What is Australia's counterpart or reference point? Is it global, Vietnam, or China? The previous sentence does not allude to any specific region for comparison.
Overall, the abstract needs to be strengthened to effectively convey the importance of this review article, particularly emphasizing the case for Australian dragon fruit.
Author Response
|
Comments 1: In general, the article is well-written and clearly articulates and highlights key points and topics. Specifically, the section “Recommendation on Australian Dragon Fruit Standard Establishment” enhances the novelty of this review by proposing a potential call to action for those involved in the production, processing, and consumption of dragon fruit.
|
|
Response 1: Thank you for the feedback. We are pleased that you found this content thorough and engaging. Including an industry-focused discussion was an intentional effort to provide a more comprehensive perspective, as it bridges the gap between scientific insights and practical applications. This review collaborates with Australian dragon fruit industry to explore more potential on establishing industry standards and eventually assist with finding the competitive advantage of Australian-grown dragon fruit as local growers suffer the price competition from imports. If you are interested in this, there is another paper from our team trying to address this issue: https://www.mdpi.com/2311-7524/10/10/1048
|
|
Comments 2: I encourage the authors to clearly state the aim of this review instead of using vague phrases like “not only introduces” as mentioned in lines 17–19.
|
|
Response 2: The expression is now changed. Please refer to the Abstract section on page 1, from line 17 to 20, with track changes.
Comments 3: Line 22: The phrase “in contrast, the Australian” is grammatically unclear. What is Australia's counterpart or reference point? Is it global, Vietnam, or China? The previous sentence does not allude to any specific region for comparison.
Response 3: This expression is now changed. Please refer to the Abstract section on page 1, from line 22 to 24, with track changes.
Comments 4: Overall, the abstract needs to be strengthened to effectively convey the importance of this review article, particularly emphasizing the case for Australian dragon fruit.
Response 4: Thank you for the feedback. The Abstract is now modified accordingly. Please find the changes from line 13 to 38 on page 1, with track changes.
|

Reviewer 4 Report
Comments and Suggestions for Authors
The article provides an extensive review that covers the most important aspects of dragon fruit. However, I have found a few errors, and I would like to offer some suggestions:
- Line 14: The authors should avoid using "etc." after "and." They should either add another country or remove "etc." entirely.
- From lines 66 to 75, there is only one reference cited. Is all this information derived from the same article?
- The figure depicting all the Hylocereus species is very nice.
- Line 111: Are there no data on the production of this fruit after 2017-2018? Six years seems like too long a gap to present an updated review on this fruit.
- Section 4.2 appears to be very comprehensive, with a table listing the main phytochemicals. However, I think the layout could be improved, especially with regard to the structure of the table. Perhaps reducing the size of some figures and allowing more space for the other columns would enhance readability. Nonetheless, I found it very well done.
- All the activities associated with this fruit are well described, beginning with a general introductory section about the type of activity (antimicrobial and antiviral, anti-inflammatory, antioxidant, anti-diabetic, anti-obesity, or anticancer), followed by related articles and properly addressed topics, although I believe some sections are somewhat lengthy. However, the section on probiotic properties could be improved. First, most of the discussion concerns the prebiotic effects (substances that promote the growth of beneficial microorganisms) of the fruit rather than probiotics (microorganisms that have a positive effect on the individual). Therefore, this section should be renamed "Prebiotic Properties," or at least "Prebiotic and Probiotic Properties." As I mentioned earlier, I think all of these sections (4.3.x) could be more concise to reduce the length of the review.
- The analysis of the industry associated with the fruit is very thorough and interesting, as such content is not always included in this type of article.
In general, this is a very good review article, although it is somewhat lengthy. Some parts, such as section 4.3, could be shortened by reducing the level of detail on general concepts like inflammation, antibacterial properties, or antioxidants. The bibliography used is highly up-to-date and carefully selected in accordance with the article.
Author Response
|
Comments 1: Line 14: The authors should avoid using "etc." after "and." They should either add another country or remove "etc." entirely.
|
|
Response 1: This is now changed, please refer to line 14 on page 1, with track changes.
|
|
Comments 2: From lines 66 to 75, there is only one reference cited. Is all this information derived from the same article?
|
|
Response 2: Only the first sentence was from that reference and I have deleted the sentence as it is not needed anymore. Please refer to line 99 on page 2.
Comments 3: The figure depicting all the Hylocereus species is very nice.
Response 3: Thanks for the comment. Yes, they look attractive and I insert them in the manuscript for better reading experience.
Comments 4: Line 111: Are there no data on the production of this fruit after 2017-2018? Six years seems like too long a gap to present an updated review on this fruit.
Response 4: Thank you for point this out. I agree that it may be outdated. However, there is no information after the 2017-2018 report which is as comprehensive as it is, involving several major production regions, especially post-covid, and that’s why I used 2017-2018 data.
Comments 5: Section 4.2 appears to be very comprehensive, with a table listing the main phytochemicals. However, I think the layout could be improved, especially with regard to the structure of the table. Perhaps reducing the size of some figures and allowing more space for the other columns would enhance readability. Nonetheless, I found it very well done.
Response 5: Thank you for the comment. As some of the compounds are very complex, I have to leave enough space for better readability. However, I made some compound structures smaller. Please refer to Table 2 on page 8, starts from line 426.
Comments 6: All the activities associated with this fruit are well described, beginning with a general introductory section about the type of activity (antimicrobial and antiviral, anti-inflammatory, antioxidant, anti-diabetic, anti-obesity, or anticancer), followed by related articles and properly addressed topics, although I believe some sections are somewhat lengthy. However, the section on probiotic properties could be improved. First, most of the discussion concerns the prebiotic effects (substances that promote the growth of beneficial microorganisms) of the fruit rather than probiotics (microorganisms that have a positive effect on the individual). Therefore, this section should be renamed "Prebiotic Properties," or at least "Prebiotic and Probiotic Properties." As I mentioned earlier, I think all of these sections (4.3.x) could be more concise to reduce the length of the review.
Response 6: Thank you for your valuable comments and suggestions. I appreciate your observation regarding the probiotic section and your recommendation to rename it as "Prebiotic Properties" or "Prebiotic and Probiotic Properties." Upon review, I acknowledge that the discussion predominantly focuses on the prebiotic effects of dragon fruit, and the section title is now updated to “Prebiotic and Probiotic Properties”. Please refer to line 975 on page 20.
Regarding your concern about the length of the 4.3.x sections, I understand your suggestion to make these sections more concise. However, the current level of detail is essential to comprehensively cover the scope of the review and provide an in-depth understanding of the various health-related activities associated with dragon fruit. Each subsection includes critical information on mechanisms, key bioactive compounds, and supporting studies, which are vital for presenting a well-rounded and thorough discussion for the audience. Reducing the length could risk omitting important details that might be relevant to readers from various disciplines. However, I took out some of the paragraphs in section 4.3 to shorten the review a bit, these changes can be found in line 492 on page 12, line 614 on page 14, line 730 and line 748 on page 16, with track changes.
Comments 7: The analysis of the industry associated with the fruit is very thorough and interesting, as such content is not always included in this type of article.
Response 7: Thank you for the comment. We are pleased that you found this content thorough and engaging. Including an industry-focused discussion was an intentional effort to provide a more comprehensive perspective, as it bridges the gap between scientific insights and practical applications. This review collaborates with Australian dragon fruit industry to explore more potential on establishing industry standards and eventually assist with finding the competitive advantage of Australian-grown dragon fruit as local growers suffer the price competition from imports. If you are interested in this, there is another paper from our team trying to address this issue: https://www.mdpi.com/2311-7524/10/10/1048
|

Round 2
Reviewer 2 Report
Comments and Suggestions for Authors
The author responded to my comment, but I still struggle to understand what kind of indicators this manuscript can provide to government agencies developing standards. Perhaps the author can try to add 3. Global Dragon Fruit Production to the standards of dragon fruit in various countries or other research. As far as I know, China has relevant standards and guidelines for dragon fruit. In addition, if the purpose is to provide information as a guide, then a literature review on the known physiological activities of dragon fruit is actually not that important. A literature review comparing the variables affecting the physiological activities of dragon fruit may be more valuable.
Author Response
|
Comments 1: The author responded to my comment, but I still struggle to understand what kind of indicators this manuscript can provide to government agencies developing standards. Perhaps the author can try to add 3. |
|
Response 1: Thank you for your insightful comments. I understand the need for clarity regarding the indicators that could assist government agencies in developing standards. In the manuscript, especially section 8 on page 27-28 from line 1286-1339, I have outlined basic quality indicators such as size, shape, and defect tolerance, which are universally recognized parameters in the horticulture industry for defining fruit quality. These indicators could serve as the foundational framework for an emerging Australian dragon fruit quality standard. These recommendations are detailed in specific sections of the manuscript, which I have highlighted in the abstract on page 1 from line 30-32, introduction on page 2 from line 120-121, section 8 Recommendation on Australian Dragon Fruit Standard Establishment on page 27-28 from line 1286-1339, conclusion section on page 29 from line 1372-1373, with track changes. They offer actionable steps that government agencies could use as a starting point to fill the current gap in standards. I hope this additional explanation clarifies how the manuscript contributes to establishing practical and comprehensive guidelines.
|
|
Comments 2: Global Dragon Fruit Production to the standards of dragon fruit in various countries or other research. As far as I know, China has relevant standards and guidelines for dragon fruit.
|
|
Response 2: Thank you for the comments. The manuscript includes a dedicated section on global dragon fruit production (Section 3, pages 5–6, lines 304–344) and global standards (Section 7, pages 25–27, lines 1199–1285). Additionally, Table 3 (page 26) summarizes the existing global standards for dragon fruit. Unfortunately, despite our efforts, we faced difficulties in accessing China's dragon fruit standards, primarily due to restricted access to official documents. We would be happy to include any references or additional information accessible to the editors, should it be required. The standards detailed in Section 7 represent those we could obtain in English and are derived from publicly available resources. While we acknowledge the importance of including China's standards, we focused on accessible documents to ensure the information provided was accurate and verifiable. Should there be future opportunities to collaborate or obtain translations of China's standards, we would be eager to integrate them into subsequent work.
Comments 3: In addition, if the purpose is to provide information as a guide, then a literature review on the known physiological activities of dragon fruit is actually not that important. A literature review comparing the variables affecting the physiological activities of dragon fruit may be more valuable.
Response 3: Thank you for this valuable suggestion. The Australian dragon fruit industry is relatively small and still in its early stages of development. As a result, this manuscript primarily aims to provide foundational baseline and insights to help establish industry sustainable standards, focusing on quality indicators and the potential inclusion of bioactive phytochemicals to enhance market appeal dedicated to fruits cultivated and curated in Australia.
While a literature review comparing the variables affecting the physiological activities of dragon fruit could indeed provide additional insights, the current review is tailored to support the growth of the Australian industry by highlighting the therapeutic potential and market value of dragon fruit. This approach serves to position Australian-grown dragon fruits in premium markets and align with global trends. We appreciate your suggestion, and future research could explore how environmental, genetic, and postharvest factors influence the physiological activities of dragon fruit. This would align with your recommendation and provide more valuable insights as the industry matures. |
